# Rebound effects undermine carbon footprint reduction potential of autonomous electric vehicles

Nuri C. Onat [1] ✉, Jafar Mandouri [1,2], Murat Kucukvar[3], Burak Sen [4], Saddam A. Abbasi [5,6], Wael Alhajyaseen [1,7], Adeeb A. Kutty [1], Rateb Jabbar [8], Marcello Contestabile[9,10] & Abdel Magid Hamouda[11]

Autonomous vehicles offer greater passenger convenience and improved fuel efficiency. However, they are likely to increase road transport activity and life cycle greenhouse emissions, due to several rebound effects. In this study, we investigate tradeoffs between improved fuel economy and rebound effects from a life-cycle perspective. Our results show that autonomy introduces an average 21.2% decrease in operation phase emissions due to improved fuel economy while manufacturing phase emissions can surge up to 40%. Recycling efforts can offset this increase, cutting emissions by 6.65 tons of Carbon dioxide equivalent per vehicle. However, when examining the entire life cycle, autonomous electric vehicles might emit 8% more greenhouse gas emissions on average compared to nonautonomous electric vehicles. To address this, we suggest; (1) cleaner and more efficient manufacturing technologies, (2) ongoing fuel efficiency improvements in autonomous driving; (3) renewable energy adoption for charging, and (4) circular economy initiatives targeting the complete life cycle.

Autonomy and electrification are two major revolutionary technologies in the transportation sector, particularly for personal private use[1]. Internal Combustion Engine vehicles (ICVs) powered by fossil fuels are conventionally the most prevalent mode of transportation and therefore, they contribute significantly to global climate change. Transportation was responsible for emitting 2.9 billion metric tons of Carbon Dioxide ($CO_2$) emissions in 2019[2]. According to the Intergovernmental Panel on Climate Change, since 1850, each of the preceding four decades has been steadily warmer than the previous

decade[3]. This sets an alarm for the changes our planet is going through and requires immediate actions to cut emissions by adopting more environmentally friendly modes of transportation with fewer emissions such as electric vehicles. Depending on the source of electricity generation, electric vehicles are likely to reduce greenhouse gas (GHG) emissions compared to ICVs[4]. Additionally, autonomous vehicle (AV) technologies are emerging and major technology companies such as Google and Tesla invest in the testing and implementation of autonomous vehicle technologies[1]. These emerging vehicles use sensors to

[1]Qatar Transportation and Traffic Safety Center, College of Engineering, Qatar University, Doha, Qatar. [2]Engineering Management, College of Engineering, Qatar University, Doha, Qatar. [3]Department of Business Ethics & Legal Studies, Daniels College of Business, University of Denver, Denver, CO, USA. [4]SAU Center for Research & Development, and Applied Research (SARGEM), Faculty of Engineering, Sakarya University, 54050 Sakarya, Turkey. [5]Statistics Program, Department of Mathematics, Statistics, and Physics, College of Arts and Sciences, Qatar University, 2713 Doha, Qatar. [6]Statistical Consulting Unit, College of Arts and Sciences, Qatar University, Doha, Qatar. [7]Department of Civil and Environmental Engineering, College of Engineering, Qatar University, Doha, Qatar. [8]The KINDI Center for Computing Research, College of Engineering, Qatar University, Doha, Qatar. [9]Qatar Environment and Energy Research Institute, Hamad Bin Khalifa University, Qatar Foundation, Doha, Qatar. [10]Imperial College London, Faculty of Natural Sciences, Centre for Environmental Policy, London, UK. [11]Department of Mechanical and Industrial Engineering, College of Engineering, Qatar University, Doha, Qatar. ✉e-mail: onatcihat@gmail.com

monitor their surroundings, artificial intelligence, and actuators for vehicle self-control to travel with little or no human interference[5]. Autonomous vehicles can enable commuters who are unable to drive, such as elderly people and those with disabilities to travel with ease. AVs introduce benefits such as driving more efficiently, reducing traffic congestion due to connected autonomous driving, and hastening the adoption of alternative fuel vehicles[6]. AVs have the potential to decrease GHG emissions mainly in a couple of ways including reduced need for parking spaces, as looking for a parking space contributes significantly to one-third of traffic within urban areas, and because of the increased traffic, vehicles are forced to stay on the road longer and use more fuel as a result which emits GHG emissions[7]. AVs could be programmed to operate more efficiently compared to human drivers, as they can drive on the most efficient routes, maintain steady speeds, and avoid unnecessary accelerations and decelerations[8]. This would reduce fuel consumption, and thus reduce GHG emissions. On the other hand, AVs could lead to an increase in GHG emissions since passengers could travel more due to easier travel[7]. It entails reaching destinations faster because of increasing capacity, fewer accidents, and cheaper travel, attracting passengers to travel more frequently[7]. Furthermore, AVs can encourage people who could not drive to travel more as they can provide people who have difficulty using traditional modes of transportation with access to a new mode of mobility[7]. As can be seen, investigating the AV effect of GHG emissions is not a trivial task and is associated with high uncertainties[7].

Considering the fuel economy benefits introduced by AVs gained by reducing congestion and lowering the magnitudes and speeds of the vehicle's acceleration and deceleration, the total environmental impacts intuitively should be less than a nonautonomous version of the same-fuel vehicle[9]. However, fully autonomous vehicles can also lead to a substantial increase in road transport activity, which would shift the balance in the opposite direction[10]. While it is not an easy task to predict how the combined effect of electrification and automation will play out overall, specific use cases have been studied in the literature that show that autonomous vehicles have the potential to lead to net GHG emission reductions[11,12]. However, these studies mainly compare non-AV fossil-fuel ICVs with electric AVs. In contrast, to understand If autonomous vehicles can lead to fewer GHG emissions, we should compare autonomous and nonautonomous versions of a same-fuel type vehicle, considering rebound effects and also including a life cycle perspective to the analysis.

Future technology adoption is assumed to follow existing technological diffusion trajectories in each sector, as evidenced by market statistics, including an increase in electric vehicle adoption[13]. In this study, we focus on electric vehicles, and we compare an autonomous electric vehicle with a nonautonomous electric vehicle by considering of different levels of GHG intensity of the electricity used for recharging, in one case based on natural gas power generation, in the other based on solar power. The consideration of these two charging scenarios also allows us to distinguish the benefits of cleaner electrification and autonomy separately. The rebound effect, described in the literature as a person's or a system's reaction to new technology and the consequences of these reactions on total resource use, is a key topic to consider in AV adoption[14]. As stated before, the introduction of AVs may lead to increased road transport activity thanks to their improved fuel efficiency and comfort[14]. Such rebound effect has repercussions not only on the use phase of the vehicle but also on the manufacturing and end-use phases, as we will explain further in "Results", hence the importance of studying the life cycle impacts of AVs as well. The rebound effect could have a direct impact on the life cycle impacts by increasing the energy usage induced by the improved fuel efficiency due to autonomy[14]. On the other hand, obscure consequences could be in the form of a change in the public's preferences when a new technology is introduced or through improved energy efficiency leading to manufacturing cost reduction resulting in a reduction in a product's price, thus encouraging additional consumption[15].

Analyzing autonomous electric vehicles from a life cycle perspective has received limited attention in the literature[16]. The rebound effect associated with AVs, although identified in early studies, has also not been extensively studied since[10]. According to our literature search in the Scopus database in Feb. 2022, most of the studies in this area were published in 2019 and onwards, which implies that this field is emerging and needs further research and assessment[9-12]. One paper accounted for the carbon footprint of autonomous vehicles and concluded that a 100% penetration rate of autonomous vehicles leads to a decrease in commuting time, as well as an enhancement in the vehicle's network's speed in terms of communication between the autonomous vehicles and the network[17]. They came up with a result of an increase in the carbon footprint of autonomous vehicles when compared to internal combustion engine vehicles, hybrid electric vehicles, and battery electric vehicles. This increase in the carbon footprint is explained by the additional manufacturing of lithium-ion batteries as well as electronic parts used in an autonomous system. This increase in manufacturing batteries is caused by the additional driving due to autonomy, necessitating replacing the battery earlier than in the case of AVs. Furthermore, more capacity is needed to power the additional electronic systems used by AVs. A shortcoming of their study is their negligence of the end-of-life potential in saving emissions by recycling the materials used in the vehicle when being salvaged. While another paper examined a real-life application of shared autonomous electric vehicles in Germany[18]. It is based on utilizing autonomous vehicles to act as driverless taxis and ride-hailing customers on the same trips. Their findings proved autonomous vehicles could reduce trip cost per kilometer by 60% when compared to privately-owned internal combustion engine vehicles by 2025 while utilizing a mix of renewable and nonrenewable energy resources for electricity generation. They have determined that autonomous vehicles can reduce commuting time since commuters will not have to look for a parking space, as well as congestion due to autonomy. In addition, in terms of energy consumption, using autonomous vehicles fleet provides significant advantages over identical internal combustion engine vehicles by up to 67%. Another study explored the use of autonomous vehicles within university campuses and showed that they could reduce greenhouse gas emissions by 36% at a 40% penetration level[19]. As the penetration level increases, there will be a greater reduction in greenhouse gas emissions. Automation may have several favorable and unfavorable effects on on-road vehicles' energy consumption and greenhouse gas emissions through changes in road transport demand, vehicle design, operational characteristics, and fuel choices[10]. Their findings indicate that high uncertainty is involved in autonomy's effect on greenhouse gas emissions, depending on which effects are prioritized, automation might alter energy use and greenhouse gas emissions from road transportation by halving them or doubling them[10].

A large research gap has been discovered in the literature as studies failed to account for uncertainties, particularly in the context of life cycle evaluations for autonomous electric cars. Using Vienna as a case study, one study explores the potential effects of autonomous vehicles on urban mobility and the environment[20]. They concluded that while AVs may boost safety, efficiency of time use, and accessibility, they may also result in undesirable trade-offs like more frequent travel and more motorized road traffic. The study also reveals that grid decarbonization, electrification of the mobility sector, and effective mass transit are crucial to lowering emissions from Vienna's transportation system. Furthermore, they determined that the degree to which these conditions are maintained will regulate the direction of GHG emission development and that AV mobility is unlikely to play a significant role. However, the research fails to conduct a thorough uncertainty-based life cycle evaluation. Furthermore, the range of the amount of trip increase attributable to the rebound effect, from 8% to

57%, reflects a high degree of uncertainty. Another study has investigated the life cycle costs and GHG emissions of various electric bus models for Singapore's public transportation system[21]. The research employed integer linear programs to optimize the trip scheduling ensuring that the size and mileage of the fleet were kept to a minimum. The optimal case highlights that GHG emissions associated with autonomous electric buses decreased by 47%, while the life cycle cost decreased by 14% compared to conventional internal combustion engine buses. One of the study's shortcomings is that it did not account for the charging stations' expenses and emissions, which could have a substantial impact on the study's conclusions. To add on, the study only looked at a small number of routes and neglected other factors such as passenger safety and comfort. Finally, one study has conducted a study to assess the environmental effects of Austin, Texas's deployment of autonomous taxi (AT) fleets[22]. The study's specific objective was to contrast three updated human-driven vehicle scenarios with five AT fleet scenarios with the current baseline scenario, which involves human-driven cars. These scenarios vary in terms of autonomy, powertrain, range, charging type, and vehicle lifetime. Their conclusions indicated that compared to human-driven vehicle scenarios, AT fleets consumed less energy and emitted less GHG emissions. The scenarios for the short-range, electric, and fully autonomous AT fleets utilized the least amount of energy and generated the fewest greenhouse gas emissions. It is vital to keep in mind, though, that the study had some limitations, such as assuming that the travel demand is constant throughout all the scenarios. Addressing the gaps in the literature requires a comprehensive approach, particularly with an emphasis on the rebound effects of AVs from a life cycle perspective.

Assessing and analyzing the carbon footprint of emerging autonomous electric vehicle technologies requires the development of integrated novel methodological approaches. As fully autonomous vehicles are currently being experimented with and on a pilot scale, quantifying and analyzing the environmental impacts of autonomous electric vehicles have certain challenges and uncertainties (please see Supplementary Tables 1–3 for components, purpose, and vehicle material composition). For instance, there are uncertainties associated with user behavior and its consequential impacts. How society would perceive technology and how their travel patterns would change in response to the improved comfort in driving and fuel efficiency. Globalization increased the complexity of the environmental assessment of products. For example, geographic location of manufacturing (e.g., manufacturing li-ion battery and extraction of raw material), design of systems, operation/use phase, and value distribution of product/system across a global complex supply chain are distributed around the world. To address these methodological challenges, we employ interdisciplinary methods drawn from decision science including system thinking practices, advanced uncertainty-based life cycle assessment (LCA), environmentally extended multi-regional input–output (MRIO) analysis, data science, and social sciences surveys. By adopting a systems thinking approach, we reveal rebound effects and behavioral aspects influencing the environmental impacts of autonomous electric vehicles.

In this work, we demonstrate the carbon footprint trade-off between autonomous and nonautonomous electric vehicles by developing a comprehensive uncertainty-based life cycle assessment approach. The fuel efficiency savings due to autonomous driving are not enough to reduce total life cycle emissions due to the increase in emissions stemming from rebound effects for both the operation and manufacturing phases. Autonomous electric vehicles might emit 8% more greenhouse gas emissions on average in total compared to nonautonomous electric vehicles. While solar charging can significantly reduce emissions for both autonomous and nonautonomous electric vehicles, further improvement and research are needed to decrease emissions due to rebound effects. To offset the increased

emissions from rebound effects, several areas of improvements are suggested; (1) cleaner and more efficient manufacturing technologies, (2) further fuel efficiency improvement in autonomous driving; (3) solar charging, (4) circular economy applications and innovative business models targeting both manufacturing, operation, and end-of-life phases. We must advance our understanding of the economic, social, and environmental impacts of emerging technologies such as autonomous electric vehicles and related global supply chain decisions. The findings of this research can be applied to policy development and strategic decision-making, such as understanding the trade-offs, challenges, risks, and sustainability impacts of adopting emerging autonomous electric vehicle technologies.

## Results
### Behavioral aspects of rebound effects: insights from the survey
To understand social aspects contributing to the rebound effects, it is important to analyze user characteristics that are more likely to increase rebound effects. There are several research efforts assessing user characteristics of autonomous cars in the literature. Studies show that people who have a high level of education have more trust in AVs and are more likely to be early adopters of AVs[23]. In addition, educated individuals have a positive opinion of AVs and are willing to pay more for them[24]. When considering the effect of age on AV adoption, with increasing degrees of automation, elderly adults were less likely to accept such technology[25,26]. For our study, based on the survey, 86% of old adults (46 years or older) are willing to adopt AVs when they become available, higher than other groups and contradicts the findings of the literature. This could be a result of cultural differences and public perceptions about such emerging technologies. Aiming to investigate our survey's user characteristics, a generalized linear model was employed to identify groups more likely to face increased rebound effects when autonomous vehicles are introduced. Findings highlight households with 3–4 adults are more likely to increase rebound effects with a 41% likelihood. We also observe that they are the group with the highest average rebound effect value of 29% compared to 22% and 20% for households with 1–2 individuals, and 5 and over individuals, respectively. Total annual traveling distance is an important parameter that amplifies the impact of rebound effects. For this reason, a generalized linear model was developed to investigate the categories likely to be traveling more annually. Those categories were identified to be full-time and part-time employees, bachelor's degree holders, and non-experienced drivers (people with 0 years of driving experience). To start, Full-time and part-time employees have jobs to attend, unlike other categories such as others. As for bachelor's degree holders, they represent over 62% of the people who travel 30,000 km or more annually. Furthermore, 54% of bachelor's degree holders travel 20,000 km or more annually, ranked first among all other degree holders.

### Carbon footprint of autonomous electric vehicles
We consider four scenarios for assessing the carbon footprint of autonomous vehicles. These Scenarios are as follows: Scenario 1: Battery electric vehicle—Electricity generated from the natural gas-fired power plant, Scenario 2: Autonomous battery electric vehicle—Electricity generated from the natural gas-fired power plant, Scenario 3: Battery electric vehicle— Electricity generated from Photovoltaics, and Scenario 4: Autonomous battery electric vehicle—Electricity generated from Photovoltaics. Figure 1 shows the Global Warming Potential (GWP) of the four scenarios considered. For the case of electricity generation using natural gas, the average life cycle GWP for nonautonomous vehicles in Scenario 1 was 31 tCO$_2$-Eq. per vehicle, while for autonomous vehicles in Scenario 2, it totals 34 tCO$_2$-Eq. per vehicle. Based on those findings, autonomy is expected to increase life cycle emissions by 8% on average due to increased travel and manufacturing of autonomous systems. This result is an average

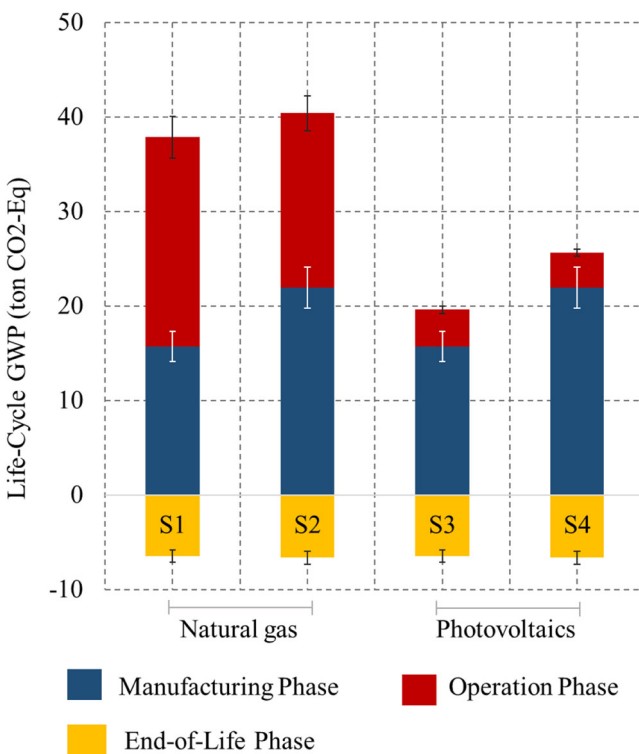

**Fig. 1 | Global warming potential results.** Data are presented as mean values of a population size of $n = 10,000$ (Bootstrapping samples) life cycle phase emissions, while error bars in the form of standard deviations for each scenario are shown to illustrate the dispersion of data points. The lower and upper bounds for each scenario are mentioned as follows (Lower bound, Upper bound); Scenario 1: Manufacturing phase: (8.66, 27.05), Operation phase: (14.69, 26.63), End-of-Life phase: (−8.32, −2.59); Scenario 2: Manufacturing phase: (11.65, 37.56), Operation phase: (12.13, 22.08), End-of-Life phase: (−8.56, −2.66); Scenario 3: Manufacturing phase: (8.66, 27.05), Operation phase: (2.06, 6.68), End-of-Life phase: (−8.32, −2.59); Scenario 4: Manufacturing phase: (11.65, 37.56), Operation phase: (1.91, 6.5), End-of-Life phase: (−8.56, −2.66). The Standard Deviation values are as follows; Scenario 1: Manufacturing phase: 4.55, Operation phase: 3.83, End-of-Life phase: 1.59; Scenario 2: Manufacturing phase: 6.4, Operation phase: 3.15, End-of-Life phase: 1.64; Scenario 3: Manufacturing phase: 4.55, Operation phase: 1.13, End-of-Life phase: 1.59; Scenario 4: Manufacturing phase: 6.4, Operation phase: 1.12, End-of-Life phase: 1.64.

for the results of the twelve car models examined in this study concerning their manufacturing, operation, and end-of-life phases. In the manufacturing phase, we considered a set of countries that can manufacture vehicles, batteries, and autonomous systems, as shown in Supplementary Tables 9 and 10. Out of these manufacturing combinations, the Chinese and Indian manufacturing sectors have relatively higher emissions. For Scenario 1 and 2, the average manufacturing phase emissions are 15.71 $tCO_2$-Eq. and 21.96 $tCO_2$-Eq. per vehicle, respectively. Thus, autonomy is likely to increase the manufacturing phase emissions by an average of 40% per vehicle. The increase in manufacturing phase emissions is due to the combined effect of manufacturing additional components for the autonomous system and the rebound effects. Starting with autonomy-purpose components, their manufacturing processes are expected to emit between 0.77 and 4.19 $tCO_2$-Eq. with an average of 2.37 $tCO_2$-Eq. per vehicle. This average increase represents approximately 11% of the average total manufacturing phase emissions. The rebound effect is responsible for emitting between 2.06 and 6.63 $tCO_2$-Eq. per vehicle and an average value of 3.87 $tCO_2$-Eq. per vehicle, representing an average contribution of 18% of the average total manufacturing phase emissions. In Scenarios 1 and 2, operation phase emissions are 22.18 $tCO_2$-Eq. and per vehicle 18.47 $tCO_2$-Eq., respectively.

Autonomy improves fuel economy[27], and thus, reduces operation phase emissions by an average of 17%.

In Scenarios 1 and 2, on average, the emission savings of 17% in the operation phase (3.71 $tCO_2$-Eq.) is not enough to offset an emission increase of 40% in the manufacturing phase (6.25 $tCO_2$-Eq.), hence autonomy results in an 8% increase in overall life cycle emissions (2.38 $tCO_2$-Eq.). We observed an increase in total life cycle emissions in ten manufacturing locations and technology. However, manufacturers from France and the USA show a decrease in total life cycle emissions. One reasonable explanation would be that the USA and France might have more efficient manufacturing technologies, relatively cleaner energy sources for manufacturing, and more efficient fuel efficiency. This finding highlights the importance of cleaner and more efficient manufacturing technologies. Consideration of upstream emissions in the manufacturing sector is important for the transition to autonomous vehicles to better utilize the manufacturing process with lesser emissions. One of the many ways to achieve this is conformance with ISO/TR 14067:2018 standard, which includes integrating the environmental aspects into the product's design and development. Additionally, material substitution or dematerialization can help reduce manufacturing-related emissions. This implies using materials that require less energy or promoting designs that require less material for manufacturing vehicle parts. An example of this is the increased usage of plastics and rubber in passenger vehicles over the years[28]. Furthermore, circular economy applications could offer a great opportunity for autonomous vehicles to reduce their emissions since their batteries require some critical metals such as Lithium and Cobalt. Cobalt has recently faced some supply chain disruptions and is becoming a challenge for manufacturing lithium-ion batteries as its demand is anticipated to increase by four times over the next four decades[29]. The market for electric passenger light-duty automobiles increased tenfold between 2011 and 2016, greatly boosting the need for lithium in the battery sector[30]. Recycling can offset some of the emissions and would decrease manufacturing emissions when compared to mining the materials all over again[31]. In the end-of-life phase, autonomous vehicles have shown average savings in emissions of 6.63 $tCO_2$-Eq. per vehicle while nonautonomous vehicles averaged a saving of 6.47 $tCO_2$-Eq. per vehicle.

Scenarios 3 and 4 demonstrate that the utilization of Photovoltaics (as an example of renewable energy) significantly reduces emissions in the operation phase. However, we still observe that autonomy increased average total life cycle emissions. The average life cycle GWP emissions for Scenarios 3 and 4 are 13.14 $tCO_2$-Eq. and 19.01 $tCO_2$-Eq. per vehicle, respectively. Using Photovoltaics in electricity generation results in a drastic decrease of 82.4% and 80% of operation phase emissions for nonautonomous electric vehicles and autonomous electric vehicles, respectively. This has resulted in manufacturing phase emissions surpassing operation phase emissions in all brands considered. So, in a scenario where we can utilize solar energy in electric autonomous vehicles, rebound effects in manufacturing phase emissions become even more important.

Figure 2 highlights a detailed breakdown of life cycle emissions for Scenarios 1 and 2. Please see Supplementary Fig. 11 for detailed breakdowns for Scenarios 3 and 4. For the electricity generation using natural gas, introducing autonomy causes an average life cycle emissions increase of 2.38 $tCO_2$-Eq. per vehicle. In the case of nonautonomous electric vehicles, the operation phase is the largest contributor to the life cycle emissions with an average contribution of 71% of total life cycle emissions compared to an average contribution of 55.2% for autonomous electric vehicles. Electricity consumption is the main source of emissions in the operation phase. In the manufacturing phase, car body manufacturing has the highest share in emissions. However, rebound effects and autonomous system manufacturing combined have emissions share as high as car body manufacturing. Increased emissions in the manufacturing phase are due to

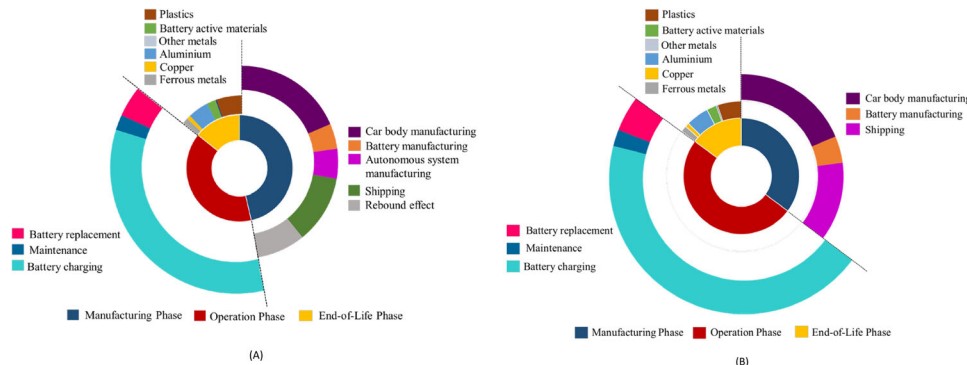

**Fig. 2 | Breakdown of life cycle emissions. A** Scenario 1 life cycle emissions breakdown. The operation phase dominates in terms of emissions, constituting the most significant portion of the life cycle emissions. **B** Scenario 2 life cycle emissions breakdown. The contribution of the operation phase to the overall life cycle emissions is reduced, closely followed by emissions from the manufacturing phase. This breakdown highlights areas for improvement by emphasizing hotspots that can be strategically addressed to achieve substantial reductions in life cycle emissions.

the rebound effect and additional manufacturing of autonomous system components contributes a combined average of 29% of manufacturing phase emissions. In the end-of-life phase, plastics and aluminum are the largest contributors to emissions saving when recycled. Recycling nonautonomous vehicle components offers an average reduction of emissions of 6.47 tCO$_2$-Eq. per vehicle while autonomous vehicles present an average saving of 6.61 tCO$_2$-Eq. This slight increase in savings is due to recycling autonomous system components.

## Stochastic results of the carbon footprint analysis

The adoption of emerging technologies has many uncertainties, and these uncertainties can result in unintended consequences due to the many unknown factors in manufacturing, operation, and end-of-life phases. Hence, we employed Monte Carlo Simulation and Multivariate sensitivity analysis to deal with these uncertainties. We employed a Monte Carlo simulation to account for various possible outcomes based on the assumptions of the inputs following probability distributions. Supplementary Table 12 shows the probability distributions followed by the inputs of this case study. Figure 3 shows the results of running 10,000 samples using Monte Carlo simulation. Both the mean and standard deviation for autonomous vehicles are higher than for nonautonomous vehicles for both energy sources. We apply a multivariate sensitivity analysis to investigate the effect of changing inputs on the life cycle GWP of autonomous and nonautonomous electric vehicles. The analysis highlights which outputs are more sensitive to model input parameters. The input data includes manufacturing phase emissions, rebound effect, fuel economy, and end-of-life phase emissions. Figure 4 shows the results of the multivariate sensitivity analysis. For Scenario 1, At low input change percentiles of less than 30%, the end-of-life phase emissions are considered the dominant process in terms of increasing life cycle GWP when compared to other variables. This is also the same case for Scenario 3 where the end-of-life phase is also the prevailing phase at such a low input percentile change rate. On the other hand, for Scenarios 2 and 4 representing autonomous vehicles, fuel economy is the dominant variable for both cases as it induces the highest change in life cycle GWP among the variables considered. At higher input percentile change values up to 60%, the fuel efficiency surpasses end-of-life phase emissions in terms of changing the life cycle GWP at the greatest amount for Scenario 1, whereas for Scenario 2 in which the rebound effect and fuel economy alternate within this interval in terms of influence on life cycle GWP when altered. When considering Scenario 3, the end-of-life emissions remain the most dominant contributors to life cycle GWP when adjusted. This is also the same case for Scenario 4 at this interval as the rebound effect remains roughly the highest influencing variable. Finally, at higher input percentile change values over 80%, the manufacturing emissions

constitute the highest influencing input on the life cycle GWP of AVs while between 60% and 80% the fuel efficiency remains dominant for Scenario 1. As for Scenario 2, the manufacturing phase emissions become the most influential variable on life cycle GWP for a change rate of over 60%. This is also the same case for Scenarios 3 and 4 where such behavior is also evident. More detailed information can be found in Section 3.4 in the Supplementary Information file and Supplementary Tables 12 and 13.

## Discussion

The results indicate that autonomy in road vehicles is anticipated to increase life cycle greenhouse gas emissions by 8% on average per vehicle. Although autonomy introduces an average 17% decrease in operation phase emissions due to improved fuel efficiency, it introduces a 40% increase in manufacturing phase emissions due to the combined effect of manufacturing autonomous system components and increased travel. Together, manufacturing autonomous system components and increased travel is responsible for an average of 18.5% of the total life cycle emissions on average. This considerable amount of increase in emissions stems from autonomous driving. Considering that the goal is to maximize the benefits of autonomous driving in terms of comfort, safety, energy efficiency, and utilization of infrastructure while reducing the environmental impacts below levels of nonautonomous vehicle emissions. This can be achieved by finding ways to mitigate increased emissions from the rebound effects. For instance, If the improvement of fuel economy due to autonomous driving could be above 48% (more than double the current fuel economy value considered in this study) compared to nonautonomous driving, the emissions from rebound effects could be eliminated and there would be no emission increase due to autonomy. Therefore, the energy efficiency of autonomous driving should be further improved. Furthermore, the manufacturing emissions of these technologies could be reduced via improved efficiency of manufacturing of critical and emission-intensive technologies such as LIDAR (Light Detection and Ranging) and RADAR (Radio waves Detection and Ranging). The use of renewable energy sources for charging as well as in the manufacturing of vehicle parts can help reduce GHG emissions overall. In addition, the use of cleaner and more resource-efficient manufacturing technologies can significantly reduce GHG emissions as the results show that a significant portion of the life cycle carbon emissions of autonomous vehicles originate from their manufacturing processes. The surge in shared AV services, like ride-sharing and carpooling, amplifies vehicle occupancy while curbing the total number of AVs on the road, thus constraining the rebound effect in independent commuting and forestalling potential surges in travel demand[32]. Moreover, employing dynamic price mechanisms such as congestion

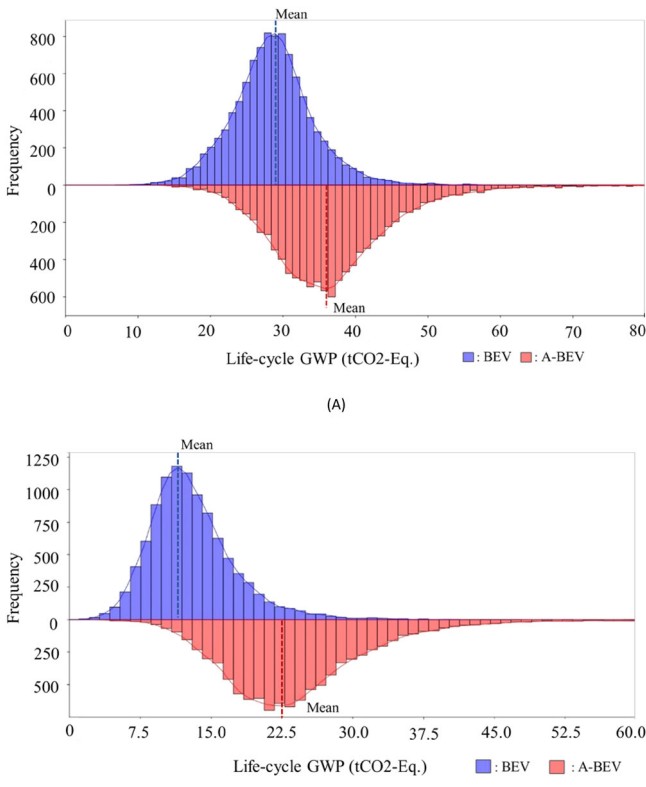

**Fig. 3 | Monte Carlo simulation results. A** Scenarios 1 and 2. **B** Scenarios 3 and 4. This analysis aids in accounting for the various sources of uncertainty within life cycle emissions of both autonomous and nonautonomous vehicles by considering numerous samples of 10,000 emissions sample values from the simulations. The two graphs provide a holistic view of the life cycle emissions of autonomous and nonautonomous vehicles by highlighting their potential difference and how they are likely to be as the simulation accounts for various uncertainties within the life cycle. %95 confidence interval on the mean are as follows: Scenario 1: (28.6, 28.8), Scenario 2: (36.6, 36.9), Scenario 3: (12.9, 13.1), Scenario 4: (24, 24.3). BEV battery electric vehicles, A-BEV autonomous battery electric vehicles. The standard deviation for S1 and S2 in (**A**) are 5.8 and 5.59, respectively, while for S3 and S4 in (**B**) they are 4.75 and 8.19, respectively.

fees dissuades needless or excessive AV utilization, fostering more eco-friendly transportation alternatives[33]. Lastly, developing strategies to transform passenger behavior away from overusing AVs and foster eco-conscious consumption patterns, while enlightening the public about the rebound effect. These captivating advertisements strive to educate people about the intricate consequences of energy-saving technologies that inadvertently augment energy usage. However, as seen in our Causal Loop Diagram (CLD) modeling, this is likely to be a complicated issue that is influenced by other several variables as discussed earlier. As autonomy is introduced extensively to private vehicles, it can contribute to a reduction in emissions through enhanced fuel efficiency as shown in balancing loop 1. However, by doing so we anticipate several unintentional effects such as stimulating an increase in travel demand (representing the rebound effect) due to enhanced driving comfort and convenience. This surge in travel demand results in amplified emissions and an intensified need for manufacturing vehicle parts, as shown in reinforcing loops 1 and 2. As can be seen, the high complexity of the rebound effect in autonomous commuting poses a significant challenge, but harnessing the potential of autonomy to reduce life cycle emissions necessitates comprehensive strategies and a holistic understanding of the dynamic interactions involved. This paper explores the intricate interactions between the variables within the life cycle of AVs in the sensitivity section where

the results offer valuable insights for policy recommendations to reduce the life cycle carbon emissions of AVs. For scenarios with input change percentiles below 30%, policies should prioritize addressing end-of-life phase emissions through recycling, reuse, and responsible disposal. In scenarios where fuel economy dominates (e.g., Scenarios 2 and 4), policies should focus on enhancing energy efficiency through lightweight materials, optimized powertrains, and eco-driving practices. As input change values increase to 60%, policies should continue to prioritize fuel efficiency improvements and address the rebound effect in shared mobility as discussed earlier in several mitigations of this phenomenon. Higher input values require policies targeting manufacturing emissions reduction through sustainable practices and low-carbon materials.

In this study, we presented a case for Qatar and rebound effects might differ regionally. Providing a comprehensive global perspective of the rebound effect requires further regionalized studies which can offer a more robust basis for country-specific decision-making in terms of autonomous vehicle adoption. Additionally, autonomous vehicle environmental impact assessment using multiple scenarios based on dynamic electricity prices and improvement in fuel efficiency can support policy development in the adoption of autonomous electric vehicles. While Global Climate Change is an urgent issue, other environmental impact categories as well as more detailed social and economic aspects of adopting autonomous electric vehicles should be studied to provide a more complete sustainability assessment.

## Methods
We utilize a range of interdisciplinary approaches rooted in decision science, such as systems thinking, advanced uncertainty-based LCA, environmentally extended MRIO analysis, data science techniques, and conducting a survey. By embracing a system's thinking perspective, we uncover how rebound effects and behavioral factors play a role in shaping the greenhouse gas emissions associated with autonomous electric vehicles.

### Understanding the rebound effects: causal loop diagramming
Autonomy introduces three main effects: (1) increased replacement frequency of vehicle parts (battery, tires, etc.) due to more vehicle use; (2) increased fuel use associated with more driving due to increased comfort that comes with autonomy and reduced travel cost per km, and (3) manufacturing of additional components of the autonomous system. To explain the dynamics linking these effects together for autonomous electric vehicles, we provide a CLD model in Fig. 5. The CLD highlights the interrelations of different variables related to competing forces (reinforcing and balancing loops) in the carbon footprint of autonomous vehicles. A reinforcing loop indicates positive causation, resulting in growth or reduction. Balancing loops indicates negative causation, attempt to move the system to a certain state, and cause a goal-seeking behavior. In the system's model, we have five reinforcing loops (R1-R5), and one balancing loop (B1).

### Reinforcing loops 1−5
R1: Autonomy → (+) Comfort of driving → (+) Travel demand → (+) Emissions →(+) Autonomy. R2: Autonomy → (+) Comfort of driving → (+) Travel demand → (+) Manufacturing of vehicle parts →(+) Emissions →(+) Autonomy in R1 and R2, Autonomous vehicles are expected to make driving easier by improved traveling comfort which would increase the travel demand[34,35]. If travel demand increases, this would result in more trips, thus more emissions[36]. Also, if the travel demand rises, this will create a need for replacing deteriorated parts due to the increased traveling, hence more emissions associated with parts manufacturing processes. The causal loop shows how improvements in the convenience of AVs can have unintended consequences, like more pollution, and how new ideas are needed to solve these problems. R3: Autonomy → (+) Fuel efficiency → (+) Travel demand → (+)

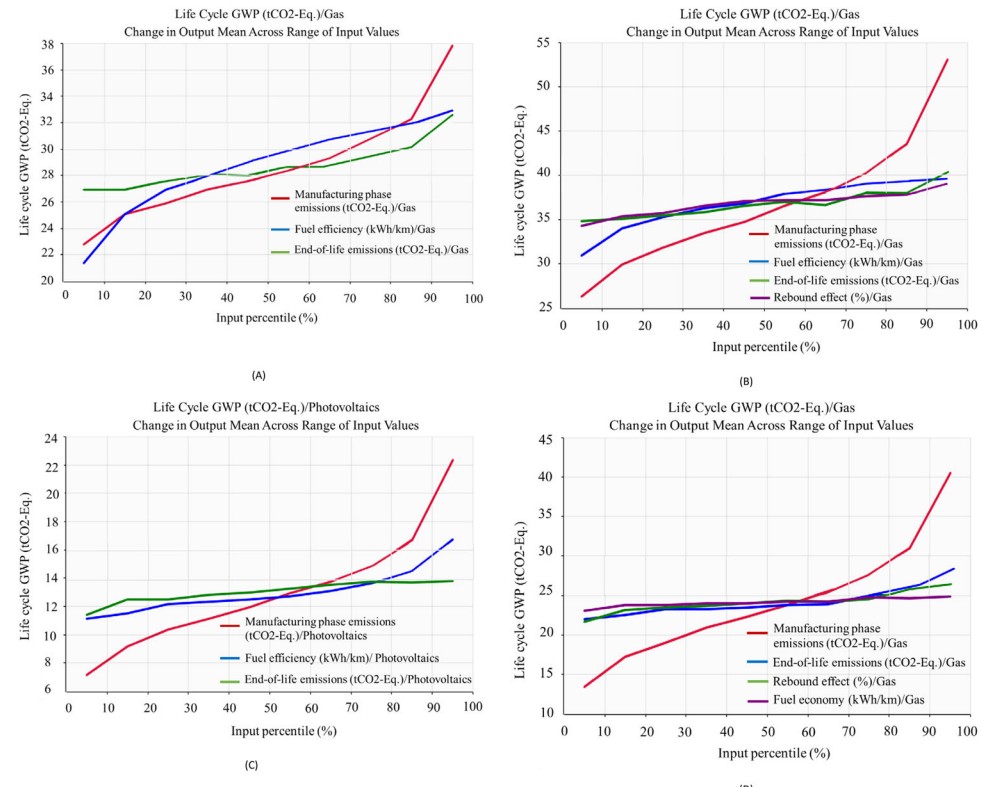

**Fig. 4 | Multivariate sensitivity analysis results. A** Scenario 1. **B** Scenario 2. **C** Scenario 3. **D** Scenario 4. This analysis supplements the results by examining life cycle input variables and how variating them would impact life cycle emissions of autonomous and nonautonomous vehicles.

Emissions →(+) Autonomy. R4: Autonomy → (+) Fuel efficiency → (+) Travel demand → (+) Manufacturing of vehicle parts → ( + ) Emissions →(+) Autonomy. When autonomy is introduced to vehicles, it is expected to enhance fuel efficiency as AVs are programmed to travel the most fuel-efficient routes and drive in the most fuel-efficient manner[9]. If there are enhancements to the fuel efficiency of a vehicle, it would result in increased travel demand as the cost of traveling per kilometer basis is reduced encouraging more people to travel[37]. In case the demand increases, it is expected that emissions and the need for replacing the deteriorated parts increase, hence more emissions. R5: Autonomy → (+) Manufacturing of vehicle parts → (+) Emissions → (+) Autonomy, If the degree of autonomy increases, it would require more components to be produced since more parts are required for the autonomous system. If the manufacturing of vehicle parts increases, it would result in an increase in emissions due to manufacturing processes.

**Balancing loop 1**
B1: Autonomy → (+) Fuel efficiency → (-) Emissions → (+) Autonomy. The balancing loop explains how autonomy could reduce emissions. If autonomy increases, this results in enhanced fuel efficiency in a unit of distance traveled (e.g., per km), which decreases GHG emissions[9].

The CLD explains the mechanism involved in the rebound effects and reveals potential intervention points to reduce GHG emissions. By adopting a systems thinking approach, we discover the rebound effects and behavioral factors that contribute to the environmental impact of autonomous electric vehicles. To further analyze the mechanisms explained in the CLD, we develop a behavioral analysis based on survey data. Our sample was collected from 330 adults based on a web-based survey questionnaire. The behavioral factors investigated include age, marital status, employment, level of education, income, number of adults per household, number of cars per household, and driving experience in years. It is important to note that this

sample may be subject to certain biases that could affect the accuracy and generalizability of our findings. One potential source of bias in our survey could be the limited availability of AVs on the market. This may result in a lack of exposure to these technologies among the population, which could lead to a psychological distance and a less nuanced understanding of their benefits and drawbacks. Through the survey, we were able to get an understanding of the user characteristics that are associated with rebound effects. We evaluated the mechanisms that are outlined in the CLD by developing a comprehensive uncertainty-based carbon footprint assessment of fully autonomous electric vehicles from a life cycle perspective.

Please see Section 2.2 in the supplementary information file for details of the generalized linear models developed by using the survey data, as well as Supplementary Table 6 which summarizes the model's results. The survey also allowed us to conduct a behavioral analysis and to understand user characteristics contributing to the rebound effects (Please see Supplementary Tables 7 and 8). To account for uncertainties about the location of the manufacturing and assembly of a potential emerging autonomous electric vehicle parts, we developed stochastic decision analysis and considered uncertainties associated geographic location of manufacturing (Supplementary Tables 9–11). For instance, we vehicle body, li-ion battery, and autonomous parts (LIDAR, RADAR, etc.) can be manufactured in different countries and assembled in another country. We investigated countries that can provide these technologies and analyzed all potential combinations of these manufacturing scenarios. Details of these calculations are presented in Section 3 in the Supplementary Information file.

**Uncertainty-based life cycle assessment**
The life cycle analysis considers all stages of the vehicle's life cycle, including its production, operation, and disposal phases by utilizing environmentally extended MRIO analysis. In addition, we investigate the possibilities of minimizing carbon emissions by recycling vehicle

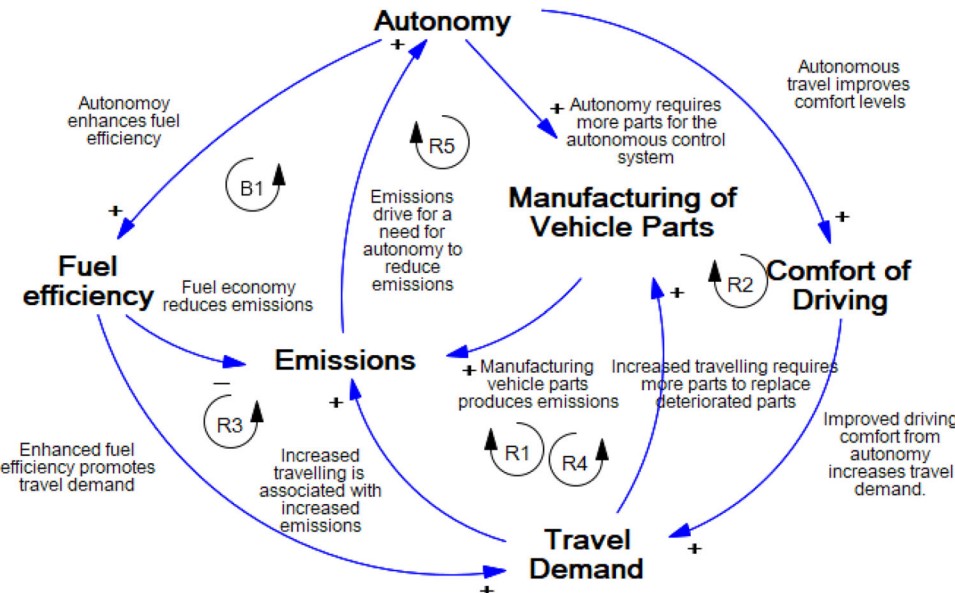

**Fig. 5 | Causal loop diagram model.** This figure represents the causal relationships between model variables explaining the rebound effects in the context of integrating autonomy into vehicles. There are four reinforcing loops contributing to an increase in life cycle emissions through increased vehicle travel and maintenance, while a balancing loop counteracts this effect by reducing life cycle emissions through enhanced fuel efficiency due to autonomy.

components to mitigate the consequences of rebound effects. Since we analyze an emerging technology that has not been fully developed, we use stochastic decision analysis to account for uncertainties associated with all life cycle phases. This analysis considers a set of potential combinations of manufacturing and assembling sedan vehicle bodies, Li-Ion batteries, and autonomous system components in nine different countries represented by twelve sedan models from the world's most renowned automobile manufacturers. The results shown in this study represent an average value of the carbon footprint for each of the twelve car brands considered. Considering multiple combinations aids in assessing the carbon footprint relevant to different choices within various aspects, such as cost, fuel efficiency, and environmental impact. Furthermore, cost combinations of vehicle bodies, Li-Ion batteries, and autonomous system components form a basis for autonomous vehicles' costs as shown in Supplementary Table 14. Similarly, we developed stochastic estimations for the carbon footprint of operation and end-of-life phases by considering potential variations in fuel efficiency and end-of-life management processes for the given twelve sedan brands. Detailed explanations about data collection and the analysis are provided in "Methods" as well as in the Supplementary Information file.

Estimating environmental impacts across global supply chains is extremely challenging with traditional process-based LCA approaches. While process-level LCA approaches are useful for the quantification of environmental impacts from a life cycle perspective, they have certain limitations when estimating emissions embedded in the supply chain of the processes in LCA due to cut-off criteria when defining the system boundary[38]. On the other hand, input–output (IO) based LCA approaches are powerful for capturing environmental impacts that are embedded in complex global supply chains, thus allowing a more holistic assessment and eliminating truncation errors due to the cut-off criteria[39]. However, IO-LCA models introduce some uncertainties due to data aggregation at a sectorial level. Input–output-based approaches are likely to yield more accurate results when estimating environmental impacts[40]. Therefore, in this research, we developed a multi-regional input–output-based LCA approach to account for indirect emissions that are embedded in the complex global supply chains of the manufacturing, operation, and end-of-life phases. The manufacturing countries are also affected by this rebound effect. For example, if autonomy increases vehicle use, more parts will be subject to depreciation and wearing, and the replacement parts will be manufactured in the country of origin, and they will induce increased environmental impacts. This impact can be seen as a supply chain-related impact. It is an indirect effect of increased use due to the rebound effect. However, this is not only for the manufacturing phase. This is true for all the supply chain-related (indirect) activities involved throughout the life cycle phases of autonomous vehicles. Therefore, we adopted the MRIO-based LCA approach to account for all the indirect effects (embedded in the global supply chains) of rebound effects. The functional unit of the analysis is per vehicle. The life cycle phases and processes involved in the analysis are presented in Fig. 6.

The integrated analysis workflow is presented in Fig. 7. After developing our hypothesis by adopting a systems thinking approach, we identified data requirements to analyze and prove the relationships defined in the CLD. We developed survey questions, collected survey data, and developed generalized linear models to analyze and understand the user characteristics and for behavioral analysis. After defining the goal and scope of the LCA, we constructed a life cycle inventory and developed the MRIO-based LCA model. We developed stochastic approaches and used methods such as Bootstrapping, Monte Carlo Simulation, and multivariate sensitivity analysis. Finally, we presented our findings using innovative data visualization techniques and provided insights, recommendations, and future work.

We collected data associated with the entire life cycle stages of an autonomous electric vehicle, including manufacturing, shipping, operation, and end-of-life phases in terms of GHG emissions and material composition. Because we developed an MRIO-based LCA model, we use producer costs as a proxy to estimate the supply chain-linked carbon footprint of processes involved in the LCA of autonomous electric vehicles. We included processes of transporting vehicle parts from different manufacturing locations to Qatar. As for the autonomous system, each component is classified into a suitable manufacturing process established in the model. The following Table 1 highlights the characteristics of the twelve car models considered in this study. For more details, refer to Supplementary Tables 10 and 11.

A list of the components and their detailed description is shown in Supplementary Table 2. We utilized the GREET model to estimate material composition in vehicles (Battery, vehicle body for quantifying

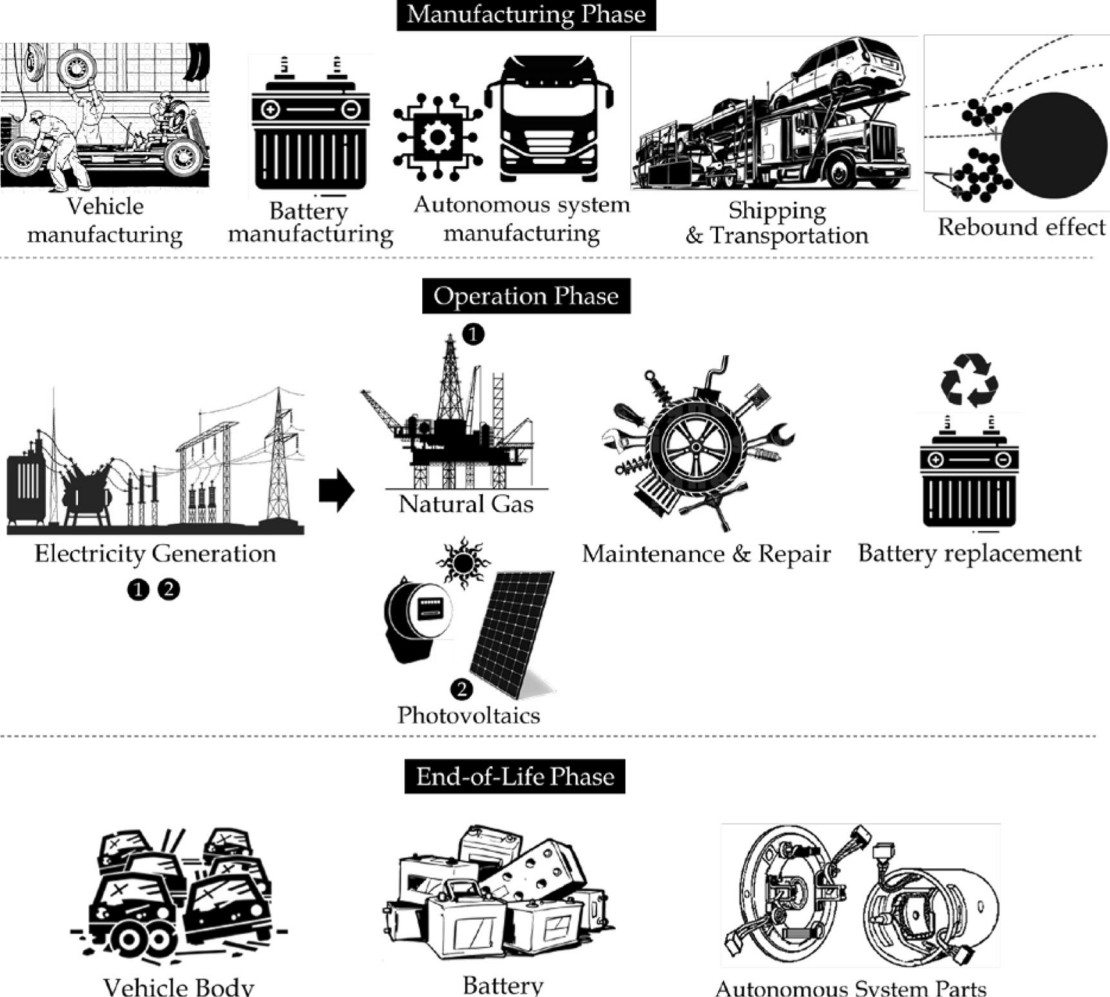

**Fig. 6 | System boundary for life cycle assessment of autonomous vehicles.** This illustration provides the scope of the study by highlighting processes within each phase of the life cycle of autonomous and nonautonomous vehicles, including their value chain. Source: Authors' work. The figure is designed by the authors combining copyright-free icons from https://www.flaticon.com. Autodesk Sketchbook, Adobe PS Suite, and Microsoft PowerPoint are used to design the figure.

the impacts of recycling in the end-of-life phase[41]). GREET model specifies all materials used in battery electric vehicles and Lithium-Ion batteries. For the autonomous system components, the material composition of each component is estimated by breaking it down into its subcomponents. Then, aggregating materials coming from the three parts to generate a list of total materials used in autonomous electric vehicles is shown in Supplementary Table 3. Afterward, we utilized the MRIO model to account for the mining emissions of each material relevant to its amount within the vehicle. As for the recycling process, Due to data absence from the MRIO model for recycling impacts for the materials used in the vehicles, we have utilized the literature for Global Warming Potential impacts associated with each of the materials used (expressed in KgCO$_2$-Eq./ Kg material recycled). Using the total materials list for each material, we accounted for the recovery rate to get the actual amount that needs to be recycled, as within recycling processes there would be some material losses. Lastly, we have accumulated the total GWP for each material present in the vehicles, to get the total recycling GWP per vehicle, which would be subtracted from the total mining GWP per vehicle to get how much the saving on emissions would be in case of recycling.

In the manufacturing phase, manufacturing of battery electric vehicles, Lithium-Ion batteries, and autonomous system components are included. Following is the shipping from their country of origin to Qatar by sea. In the operation phase, charging, maintenance, and battery replacement resemble emissions sources. Maintenance is based on real-world data collected from a public transportation company in Qatar, which operates electric trucks. The maintenance costs were scaled down to accommodate electric vehicles based on the proportion of bus price to car price. As for battery replacement, we assume that electric vehicles will require one battery replacement during their entire life cycle. This is mainly due to the rebound effect causing increased annual traveling distances thus having a greater need for the deteriorating Li-Ion battery. Lastly, for the end-of-life phase, the savings in emissions due to recycling are quantified by calculating the difference between emissions associated with mining the materials and recycling them. For further details about the calculation steps please see Section 3 in the Supplementary Information file.

## Multi-regional input−output model

MRIO Modeling is an economic technique that keeps track of money transfers among the main economic sectors across major economies in the world. Environmentally extended MRIO modeling allows tracking and analyzing resource flows by incorporating information from the National Footprint Databases and Biocapacity Accounts. MRIO models allow for tracking of resource flows between a country's primary economic sectors and satellite national accounts and breaking down national Footprint data into more targeted consumption- and industry-related components[42]. Global financial transactions between

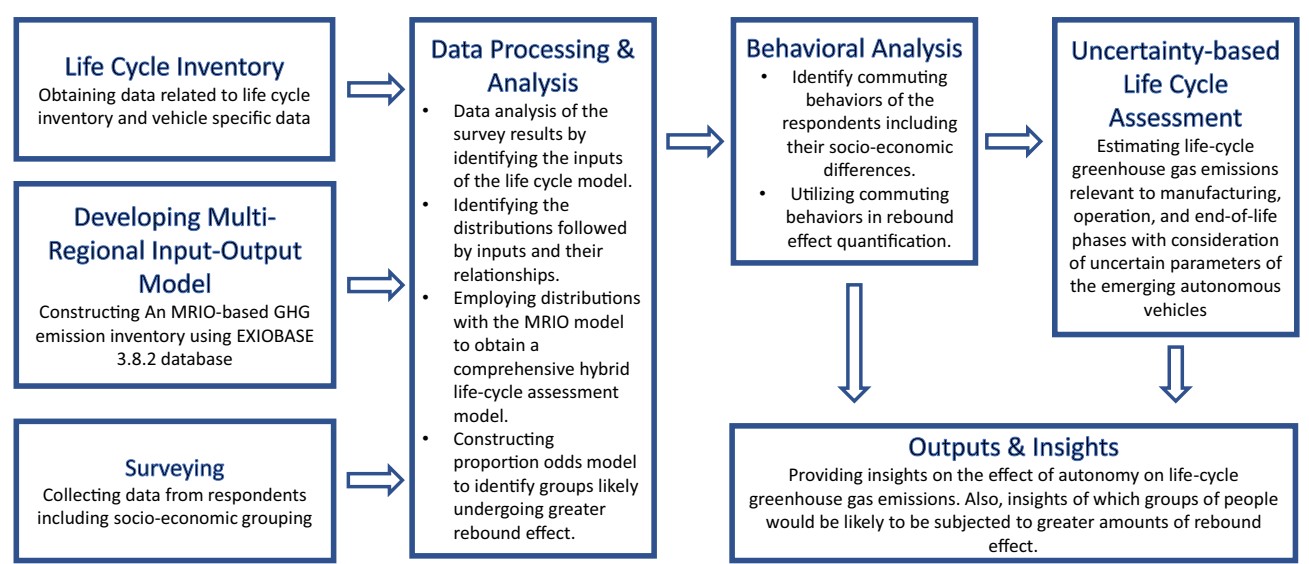

**Fig. 7 | Analysis Workflow.** This figure summarizes the methodology developed in this study along with a workflow explaining each methodological step briefly. GHG greenhouse gas, MRIO multi-regional input−output, GHG Greenhouse gas, MRIO multi-regional input−output.

governments and regions can be captured using MRIO models because of this, MRIO analysis can help to analyze how economic activity in certain sectors and countries can cause environmental impacts with consideration of global supply chains[43]. Constructing the MRIO model using Leontief's equation in Eq. (1) to access implications associated with a specific sector's output unit as well as unintended consequences of the industry's international supply networks.

$$X = (I - A)^{-1} \times y \qquad (1)$$

Where X: Total output column vector (M€); I: identity matrix; A: Coefficient matrix (M€/M€); y: total demand column vector (M€).

Next, Eq. (2) is used to obtain multipliers indicating the impact of a given sector's input of one million Euros.

$$B = E \times (DIAG(X)) - 1 \qquad (2)$$

Where B: Matrix of intensities (Per M€); E: Environmental satellite accounts.

Finally, Eq. (3) is as follows:

$$r = BLy \qquad (3)$$

Where r: total output of each sector vector; B: intensity matrix per unit of output; L: total requirements matrix.

In this study, EXIOBASE 3.8.2 is used as the MRIO database[44]. The Supply and Use Tables are used to create EXIOBASE's MRIO datasets, which comprise national and global input−output tables as well as raw data from the UN's System of National Accounts, Comtrade, and Eurostat databases, all at current prices with a steady product sales assumption. The environmental impacts considered in this study are emissions of Carbon dioxide ($CO_2$), Methane ($CH_4$), and Nitrogen dioxide ($N_2O$). Then, using these emissions, GWP is estimated based on a 100-year time series as per IPCC's 6th assessment report[45]. GWP is a measure of a greenhouse gas's ability to trap energy over a given time. This amount of energy absorbed by a ton of a specific GHG is compared to a ton of $CO_2$'s ability to trap energy.

## Surveying
The survey is confirmed to be in full compliance with all relevant ethical regulations. The study protocol has been reviewed and approved by the Ethical Committee at Qatar University, approval number: QU-IRB 1111-E/19. All participants in this survey consented to the following statement: "This questionnaire aims to investigate the perception of people about autonomous cars, mainly to understand the public's concerns, opinions, and preferences about this emerging technology. This questionnaire targets individuals who are older than 18 years old and own a Qatari driving license. All given responses will be treated with the utmost confidentiality. The results will be only used for research purposes and no attempt will be made to identify any individual or organization in any publication. Furthermore, the study is approved by the Qatar University Institutional Review Board with the approval number [QU-IRB 1111-E/19]. If you have any questions related to the ethical compliance of the study, you may contact them at QU-IRB@qu.edu.qa. The questionnaire will take around 20 min to complete. You can withdraw and stop completing the survey at any time you wish". We Built a web-based survey questionnaire utilizing the Qualtrics platform in Arabic and English languages. Information pamphlets that were disseminated through various networking groups and social media platforms included the questionnaire. In total, 589 respondents filled out the survey. Data is collected from respondents including socio-economic characteristics, AV-related questions, and commuting behaviors. Socio-economic characteristics include age, education, marital status, and employment. Supplementary Table 4 summarizes the demographics of the survey's respondents while Supplementary Fig. 1 shows the Covariate variable characteristics of respondents. AV-related information, including respondents' background knowledge of AVs and willingness to switch to AVs when they are available, was collected. While the implications, benefits, technological advancements, and technological challenges of autonomous vehicle cars are frequently discussed, the public's acceptance and perception of these vehicles received less attention[46]. For this purpose, some questions in this part were about stating opinions about the potential benefits of autonomous vehicles. The benefits include reducing congestion on roadways, reducing fuel consumption, reducing travel time, and reducing parking costs. Respondents were asked to choose one statement to express how confident they are that autonomous vehicles will bring this benefit. A summary of the perceived benefits of the respondents of autonomous vehicles is shown in Supplementary Table 7. Similarly, concerns regarding autonomous vehicles are also addressed. Concerns consist of autonomous system hacker attacks, accidents between nonautonomous vehicles and

**Table 1 | Vehicle model specifications**

| Brand reference name | Fuel efficiency (kWh/km) | Battery capacity (kWh) | Car body production cost ($) | Battery production cost ($) | Total autonomous system production cost ($) |
|---|---|---|---|---|---|
| China | 0.14 | 53.1 | 24,671 | 4906 | 6108 |
| Japan 1 | 0.19 | 71.4 | 21,482 | 6598 | 8784 |
| Japan 2 | 0.174 | 62 | 16,142 | 5729 | 8487 |
| India | 0.0968 | 30.2 | 14,482 | 2790 | 5923 |
| Korea | 0.186 | 58 | 26,210 | 5359 | 8668 |
| US 1 | 0.2 | 98.7 | 23,894 | 9120 | 7202 |
| US 2 | 0.17 | 82 | 30,893 | 7577 | 10,198 |
| Mexico | 0.1125 | 18 | 21,042 | 1663 | 6656 |
| Germany 1 | 0.182 | 55 | 28,170 | 5082 | 13,402 |
| Germany 2 | 0.172 | 83.9 | 42,829 | 7753 | 10,387 |
| Turkey | 0.18 | 90 | 26,018 | 8316 | 8420 |
| France | 0.165 | 54.7 | 22,251 | 5054 | 8266 |

This table summarizes the critical specifications for the 12 brands considered in this study. These specifications form the basis of our analysis and could be utilized to produce similar studies.

autonomous vehicles, increases in maintenance costs for autonomous vehicles, and autonomous vehicles' performance in harsh environments. Summary of the concerns is summarized in Supplementary Table 8. A full list of the questions answered by respondents is provided in the Supplementary information in Section 2. Lastly, commuting behavior variables include annual traveling distance per adult, number of cars in the household, and change in annual traveling distances in the case of autonomous vehicle adoption. The change in annual traveling distance is explained in Eq. (4) for rebound effect computation.

$$RE = \frac{\left(\frac{MVM}{GATDSA}\right) - \left(\frac{MVM}{GRAATDSA}\right)}{\left(\frac{MVM}{GATDSA}\right)} \tag{4}$$

Where RE: Rebound effect; MVM: Maximum vehicle milage; GATDSA: Generated annual traveling distances samples average; GRAATDSA: Generated rebound-adjusted annual traveling distances samples average.

Commuting behavior variables are utilized in quantifying total anticipated traveling distances with autonomous vehicles. Equation (5) highlights calculations for average rebound-adjusted annual traveling distance:

$$ARAATD = (ATD)Avg \times (RE)Avg \tag{5}$$

Where ARAATD: Average rebound-adjusted annual traveling distance; ATD: annual traveling distance; RE: The rebound effect.

**Data processing and analysis**

The screening process eliminated 9 respondents younger than 18 years, 145 respondents due to omitting their income, 104 respondents with missing data on how much their travel would increase annually if they switched to AVs, and one respondent with missing AV background knowledge data. In total, 330 respondents were used as the sample. One important variable is the average rebound-adjusted annual traveling distance, which is obtained from Bootstrapping as the sample size was relatively low. Bootstrapping is a test that uses random sampling with replacement and falls within the larger category of resampling techniques. To build the sample distribution for the desired estimate, sampling with replacement is performed[47]. This procedure is explained in Section 2.1 in SI where Supplementary Figs. 2–10 and Supplementary Table 5 show the results of this procedure. Both average annual traveling distances and rebound-adjusted annual traveling distances sets are used for quantifying how soon will we need to replace autonomous vehicles when compared to

nonautonomous vehicles. Since autonomous vehicles are anticipated to increase traveling, vehicles will deteriorate faster, and users will have to buy a new car sooner than when compared to nonautonomous vehicles. This decrease in service life is expressed as a percentage increase in manufacturing phase emissions, it is also expressed as the rebound effect used in Eqs. (4) and (5). After obtaining life cycle phase emissions for the twelve manufacturing location combinations, it is important to find the best-fit distribution of the data. This is done using @Risk software's distribution fitting feature. Outputs of this process include the distribution followed by manufacturing phase emissions, fuel efficiency, rebound effect, and end-of-life phase emissions, which are shown in Supplementary Table 12. We integrate the distributions identified in data processing & analysis and run a set of Monte Carlo simulations to account for potential uncertainties and provide stochastic estimations for the carbon footprint of autonomous electric vehicles. In addition, a multivariate sensitivity analysis is conducted to understand how sensitive the input parameters such as rebound effect, fuel efficiency, end-of-life phase emissions, and manufacturing-related emissions.

**Behavioral analysis**

To understand the relationship between the rebound effects and socio-economic characteristics, we conducted a behavioral analysis. Using the proportional odds model, an investigation is carried out to identify which groups within different socio-economic factors are more likely to travel greater distances when autonomous vehicles are introduced. The proportion odds model is a statistical analysis approach that models the relationship between an ordinal response variable and one or more explanatory variables[48]. In addition, two more models are constructed to identify groups with a higher likelihood of having better background knowledge of autonomous vehicles, as well as identifying groups traveling more on an annual basis compared to other groups. The three models share the independent variables including age, marital status, employment, level of education, income, number of adults per household, number of cars per household, and driving experience in years. The results are shown in Supplementary Table 6.

**Reporting summary**

Further information on research design is available in the Nature Portfolio Reporting Summary linked to this article.

## Data availability

Life cycle inventory and vehicle specification data available in the Supplementary Information File Survey data are available at

https://figshare.com/s/d19a396a3b540d22e88a. Exiobase 3.8.2 data are available at https://zenodo.org/record/5589597. Greet Model 2 (2022) data are available at https://greet.es.anl.gov/greet/versions.html. Source data are provided with this paper.

## Code availability

Python code is available at the following link https://doi.org/10.5281/zenodo.8327297.

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

## Acknowledgements

We acknowledge Marubeni Corporation for funding this project, under project number QUEX-CENG-MJF-EV-18/19. The collection of survey data is supported by Khatib & Alami Engineering Consolidated Engineering Company. We also thank Qatar University for the Graduate Assistantship program for funding. Open Access funding is provided by the Qatar National Library. We extend our gratitude to the funding organizations and our stakeholders who made this publication possible. We express our sincere gratitude to the anonymous reviewers for their invaluable insights and constructive comments, which have greatly enhanced the scientific rigor of our work. In addition, we extend our appreciation to all anonymous survey respondents for their cooperation and diligent participation, which significantly contributed to the quality of our research.

## Author contributions

N.C.O.: conceptualization, research methodology development, initial and final draft writing and editing, supervision, and validation. J.M.: data analysis, data collection, initial & final draft writing, literature review, and data visualization. M.K.: life cycle assessment methodology design, research methodology development, and data collection. B.S.: data analysis, research methodology development, and data collection. S.A.: statistical consulting, data analysis & curation, and data validation. W.A.: survey data collection, survey data processing, and reviewing. A.K.: data visualization. R.J.: model development, data visualization, and data collection. M.C.: draft editing and peer reviewing. A.M.H.: funding acquisition, project management, and supervision.

## Competing interests

The authors declare no competing interests.
