## [Peer Review File · Nature Communications]

REVIEWER COMMENTS

Reviewer #1 (Remarks to the Author):

Autonomous driving is a driving mode of cars based on computer technology and artificial intelligence. It is interesting and meaningful to explore the potential for reducing the carbon footprint of autonomous vehicles throughout their life cycle. The authors studied the trade-offs between improving fuel efficiency, changes in travel behavior, and increased efficiency from a life cycle perspective, and proposed measures to offset the rebound effect of carbon emissions. However, in my view, there are still some critical issues that need to be addressed.

1. The authors studied electric vehicles, but the use of fuel efficiency in the abstract could create some ambiguity. Additionally, the authors also investigated the benefits of the carbon footprint resulting from car recycling, but this was not reflected in the abstract.
2. In this study, the authors investigated the potential for reducing the carbon footprint of autonomous vehicles. However, the background section lacks an analysis of the impact of autonomous driving on greenhouse gas emissions. Please reorganize this section after accurately understanding the relationship between autonomous vehicles and carbon footprint, the current state is unacceptable.
3. An analysis of greenhouse gas emissions from autonomous and non-autonomous vehicles with the same fuel type is needed. However, the combined impact of autonomy and electrification in transportation is intuitively greater and is also a future trend. Why did the authors not study this? If automation may result in additional carbon emissions, what is the situation with electrification added?
4. The authors conducted a detailed analysis of three papers on autonomous vehicles. What is the representativeness of these papers? The authors need to read more literature for a comprehensive evaluation.
5. The sample studied was collected from 330 adults through a web-based survey questionnaire. The sample size of the study is very small. How does this justify the findings?
6. Lines 315 to 319, the authors first described the results of Figure 5, and then mentioned the carbon footprint of autonomous and non-autonomous electric vehicles when using solar energy. In fact, the authors' Fig. 5 studied the Scenario 2. The authors' analysis of this is very confusing.
7. The content studied in this paper is the carbon footprint reduction potential of autonomous electric vehicles, and there is no evaluation of the economic feasibility of their entire life cycle. Why does the Supplementary Information only provide information on the prices of cars and components, and not a specific list of materials, energy, and emissions used by the cars and their components?
8. The Table 2 in the SI file, only provide a detailed description of autonomous vehicles, and does not provide a list of materials, energy, and pollution emissions involved.
9. How is the carbon emissions generated in the manufacturing process of components and materials not included in the existing GREET database determined?

10. The supporting information only investigated the rebound effect in one region of Qatar through a questionnaire. However, in the main text, the authors compared nine countries without indicating whether these countries also experienced a rebound effect.
11. Line 456, the reviewer did not find a comprehensive list of total materials used in autonomous electric vehicles in both the main text and supplementary information. Please explain and supplement this information.
12. There is no specific list in the supporting information for the manufacturing of the vehicle body, lithium-ion batteries, and autonomous system components. How is the overall carbon emissions of autonomous vehicles calculated?
13. How are carbon emissions calculated during the autonomous vehicle usage phase? Specifically, what parts of the battery usage are included? In line 459 of the supporting information, why is the carbon emissions calculation of electricity generation using the electricity price?
14. Autonomous vehicles also consume corresponding materials and energy and produce pollution emissions during the recycling. How did the authors handle the carbon emissions generated from this part in their calculations?
15. There are some issues with the reference citations in the main text and supporting information.
16. The source of the data in Table 4 of the supporting information is unknown.
17. Lines 315 to 318 of the supporting information, the relevant calculations mainly focus on prices, and the average values of technology, raw materials, and energy for the nine countries are not reflected.

Reviewer #2 (Remarks to the Author):

The manuscript Rebound Effects Undermine Carbon Footprint Reduction Potential of Autonomous Electric Vehicles presents comprehensive analysis. However, manuscript can only be published after following major revisions.

1. There are numerous places where references are missing.
2. The authors used GREET model to estimate material composition in vehicles. The authors need to explain the suitability of the GREET model for other nations, it is important to consider the model's

applicability to different contexts. The GREET model was initially developed for the United States and focuses on vehicle technologies and energy pathways specific to the U.S. Therefore, when applying the GREET model to other nations, it may require adjustments and customization to accurately reflect the local energy mix, transportation infrastructure, and vehicle characteristics.

3. The manuscript needs better organisation. A lot of important things such as data sources, uncertainty analysis are provided in supplementary information. A part of that information in the form of tables or figures should be included in the the main manuscript.

Reviewer #3 (Remarks to the Author):

The paper provides a well-defined and impactful objectives. It investigated tradeoffs between improved fuel economy and rebound effects from a life-cycle perspective.

Several comments are provided:

- The life cycle operations with quantitative results obtained, the calculations and scope is based on a product, local, regional or global study, further supplement would allow readers with a clearer understanding on the paper.

- About the rebound effects and whether AVs have fewer emissions, what is the theoretical support for the causal loop diagram model (CLD) proposed in lines 132-138 of the text? What are the model's design sources in the absence of relevant literature support? Line 153 of the text reinforces loops4 and mentions expected fuel efficiency improvements. Is there any supporting data? Line 170 mentions rebound effects and behavioural factors leading to the environmental impacts of autonomous electric vehicles. Can a brief list of the influencing factors be given?

- About the carbon footprint of EV, four scenarios from production to use to end of life according to the LCA assessment methodology are provided. This gives the conclusions of the analysis. However, line 235 refers to 12 vehicle types, which ones are they, and whether the LCA was carried out on all the vehicle components or whether it is suggested that this section could be described in more detail. Line 274 refers to renewable energy, whereas renewable energy should refer to Scenario 3 and Photovoltaics in Scenario 3 and 4. It is suggested that this be clarified.

- The discussion section describes the results of the analysis of the entire study. Furthermore, based on the results, suggestions are made for the future development of AVs. However, the article's title considers the future development of Avs from the perspective of the rebound effect, and the discussion

section should suggest ways to mitigate the increase in carbon emissions due to the rebound effect. The CLD cycle model mentioned in the previous section should also be discussed as appropriate. The sensitivity analysis results are mentioned in line 349, but more details of the results could be presented.

- In the questionnaire studying the behaviour. Further details could be provided including the source of the questionnaire, reliability analysis (calculating Cronbach's alpha), and validity analysis results of the questionnaire.

Overall, the study is comprehensive and impactful with novelty in the study. The study analyses the rebound effect that reduces the carbon footprint of self-driving electric vehicles, regarding CLD cycle models, LCA analysis, and questionnaire analysis. Minor revision for the paper would be needed for further acceptance consideration of the paper.

Detailed Response to Reviewers' Comments

The authors extend their utmost gratitude to the reviewers and the editor for their constructive and insightful comments. In accordance with their suggestions, the authors provided detailed explanations for every concern and comment raised by the reviewers, incorporating them into the manuscript. The changes are also highlighted in the revised manuscript for the reviewers' information. In this document, the authors' responses are shown in blue color.

Reviewers and/or Editors' comments:

Reviewer #1:

Autonomous driving is a driving mode of cars based on computer technology and artificial intelligence. It is interesting and meaningful to explore the potential for reducing the carbon footprint of autonomous vehicles throughout their life cycle. The authors studied the trade-offs between improving fuel efficiency, changes in travel behavior, and increased efficiency from a life cycle perspective, and proposed measures to offset the rebound effect of carbon emissions. However, in my view, there are still some critical issues that need to be addressed.

1. The authors studied electric vehicles, but the use of fuel efficiency in the abstract could create some ambiguity. Additionally, the authors also investigated the benefits of the carbon footprint resulting from car recycling, but this was not reflected in the abstract.

The inclusion of fuel efficiency in the abstract serves to present one of the highly anticipated advantages associated with autonomous vehicles, namely their enhanced fuel efficiency. While electric vehicles do not rely on conventional fuels, their efficiency in energy consumption remains a crucial factor affecting their overall carbon emissions throughout their lifecycle. Therefore, the mention of fuel efficiency in the abstract aims to highlight the potential of autonomous vehicles in effectively addressing a substantial portion of their life cycle emissions by virtue of their fuel efficiency. Regarding recycling, per suggestion, the abstract has been revised to highlight recycling benefits in reducing carbon emissions over the entire life cycle as follows: "Vehicle recycling is anticipated to reduce life cycle emissions by 6.65 tCO₂-Eq. per vehicle".

2. In this study, the authors investigated the potential for reducing the carbon footprint of autonomous vehicles. However, the background section lacks an analysis of the impact of autonomous driving on greenhouse gas emissions. Please reorganize this section after accurately understanding the relationship between autonomous vehicles and carbon footprint, the current state is unacceptable.

We understand the importance of accurately portraying the intricate relationship between autonomous vehicles and their potential effects on greenhouse gas emissions. In response to this concern, we reorganized and enhanced the background section to provide a comprehensive analysis of this relationship. The change reflected in the background section of manuscript is as follows: "AVs have the potential to decrease GHG emissions mainly by a couple of ways including reduced need for parking spaces, as looking for a parking space contributes significantly to one-third of traffic within urban areas and because of the increased traffic, vehicles are forced to stay on the road longer and uses more fuel as a result which emits GHG emissions. Additionally, AVs could be programmed to operate more efficiently compared to human drivers, as they can drive on the most efficient routes, maintain steady speeds, and avoid unnecessary accelerations and decelerations. This would reduce fuel consumption, and thus reduce GHG emissions. On the other hand, AVs could lead to an increase in GHG emissions since passengers could travel more due to easier travel⁸. It entails reaching destinations faster because of increasing capacity, fewer accidents, and cheaper travel, attracting passengers to travel more frequently. Furthermore, AVs could encourage people who could not drive to travel more as they can provide people who have difficulty using traditional modes of transportation with access to a new mode of mobility. As can be seen, investigating AVs effect of GHG emissions is not a trivial task and is associated with high uncertainties."

3. An analysis of greenhouse gas emissions from autonomous and non-autonomous vehicles with the same fuel type is needed. However, the combined impact of autonomy and electrification in transportation is intuitively greater and is also a future trend. Why did the authors not study this? If automation may result in additional carbon emissions, what is the situation with electrification added?

The combined impact of autonomy and electrification have been studied in the literature by some of the authors in this study:

Ercan, T., Onat, N. C., Keya, N., Tatari, O., Eluru, N., & Kucukvar, M. (2022). Autonomous electric vehicles can reduce carbon emissions and air pollution in cities. Transportation Research Part D: Transport and Environment, 112, 103472

Sen, B., Kucukvar, M., Onat, N. C., & Tatari, O. (2020). Life cycle sustainability assessment of autonomous heavy-duty trucks. Journal of Industrial Ecology, 24(1), 149-164

While it is true that the impact of introducing both autonomy and electrification is anticipated to be significant in the transportation sector, it also causes an ambiguity when these two factors are analyzed in a combined way.: One could ask: Is it autonomy or electrification reducing the emissions? Some of the authors also analyzed effect of electrification on carbon footprint emissions extensively:

Onat, N. C., Kucukvar, M., & Tatari, O. (2015). Conventional, hybrid, plug-in hybrid or electric vehicles? State-based comparative carbon and energy footprint analysis in the United States. Applied Energy, 150, 36-49.

Onat, N. C., & Kucukvar, M. (2022). A systematic review on sustainability assessment of electric vehicles: Knowledge gaps and future perspectives. Environmental Impact Assessment Review, 97, 106867.

Electrification is likely to reduce emissions from transportation. However, this potential is determined by the source of electricity generation. On the other hand, effect of autonomy is not comprehensively studied. Therefore, in this study, we solely focused on comprehensive effects of autonomous driving. Our objective is evaluating the potential of autonomy in isolation, without considering the combined impact of both autonomy and electrification. This stems from the need to study the rebound effect closely, without having it being influenced by the electrification process. By doing so, we intend to provide novel insights and contribute to the existing body of knowledge in the literature as it lacks studies with this focus specifically on the rebound effect.

4. The authors conducted a detailed analysis of three papers on autonomous vehicles. What is the representativeness of these papers? The authors need to read more literature for a comprehensive evaluation.

We appreciate your concern about the representativeness of the chosen papers. We want to reassure you that we have extensively altered the literature review section, taking the representativeness into consideration. We made a systematic literature search to ensure a thorough review of the autonomous vehicle research sector by incorporating new studies. This was mentioned in the manuscript as follows: "According to our literature search in Scopus database in Feb. 2022 using keywords mentioned following the abstract", where our keywords are as follows: Carbon Footprint; Life cycle assessment; Autonomous electric vehicles; Rebound effects. In this revised version, three more studies were carefully chosen to provide a deeper understanding of the literature's findings. These additional studies were chosen for their relevance, objectivity, and importance to the field of study. We reviewed these articles to ensure that they are relevant to the study's objectives and provide useful information. These improvements, we believe, have considerably improved the literature review section, addressing your concerns regarding representativeness and assuring a more in-depth treatment of the issue.

5. The sample studied was collected from 330 adults through a web-based survey questionnaire. The sample size of the study is very small. How does this justify the findings?

The population we want to investigate include only light-duty vehicle licensed drivers. For the period 2018-2022, about 765,600 licenses of this category were either issued or renewed (we took the five years period as driving licenses in Qatar are valid for only five years). This estimation included building an Exponential Triple Smoothing

forecasting model to forecast the number of renewal and issuance of licenses in 2021 and 2022 as the official published data is limited to 2020 (CHAPTER X TRANSPORT AND COMMUNICATIONS STATISTICS, n.d.). we calculate the required sample size using the following formula mentioned in several statistics books including (Verma & Verma, 2020):

$$N = Z(\alpha/2)^2 * p*(1-p) / MOE^2,$$

- $Z_{\alpha/2}$: the critical value of the Normal distribution at $\alpha/2$ (at the current confidence level of 90%, α is 0.1 and the critical value is 1.645)
- N: Sample size
- MOE: the margin of error (5% assumed value)
- P: the sample proportion (50% assumed as recommended by default)
- N: the population size (765600)

Using this formula, we get the required sample size to be 271. Therefore, the sample size of 330 is representative of the population we aim to investigate since it is larger than the minimum required sample size.

6. Lines 315 to 319, the authors first described the results of Figure 5, and then mentioned the carbon footprint of autonomous and non-autonomous electric vehicles when using solar energy. In fact, the authors' Fig. 5 studied the Scenario 2. The authors' analysis of this is very confusing.

We apologize for the inconvenience caused by this issue. The text in the paragraph was meant to explain the multivariate sensitivity analysis results shown in Figure 5 instead of mentioning utilizing solar energy in assessing the carbon footprint of AVs. We have revised this part and ensured it now conveys the information it must do, including investigating which variables within the life cycle of AVs have the greatest potential on influencing their life cycle GWP when altered. This was mentioned in the results of the manuscript as follows: "Fig. 5 shows the results of multivariate sensitivity analysis where it highlights the extent of the impact the shown variables have on life cycle GWP of AVs. For Scenario 1, At low input change percentiles less than 30%, the end-of-life phase emissions are considered as the dominant process in terms of increasing life cycle GWP when compared to other variables. This is also the same case for Scenario 3 where the end-of-life phase is also the prevailing phase at such a low input percentile change rate. On the other hand, for Scenarios 2 and 4 representing autonomous vehicles, fuel economy is the dominant variables for both case as it induces the highest change on life cycle GWP among the variables considered. At higher input percentile change values up to 60%, the fuel efficiency surpasses end-of-life phase emissions in terms of changing the life cycle GWP at the greatest amount for Scenario 1, whereas for Scenario 2 in which the rebound effect and fuel economy alternate within this interval in terms of influence on life cycle GWP when altered. When considering Scenario 3, the end-of-life emissions remain as the most dominant contributors to life cycle GWP when adjusted. This is also the same case for Scenario 4 at this interval as the rebound effect remains roughly the highest influencing variable. Finally, at higher input percentile change values over 80%, the manufacturing emissions constitute the highest influencing input on the life cycle GWP of AVs while between 60% and 80% the fuel efficiency remains dominant for Scenario 1. As for Scenario 2, the manufacturing phase emissions become the most influential variable on life cycle GWP for a change rate of over 60%. This is also the same case for Scenarios 3 and 4 where such behavior is also evident."

7. The content studied in this paper is the carbon footprint reduction potential of autonomous electric vehicles, and there is no evaluation of the economic feasibility of their entire life cycle. Why does the Supplementary Information only provide information on the prices of cars and components, and not a specific list of materials, energy, and emissions used by the cars and their components?

Thank you for your valuable feedback. The information provided in the Supplementary Information file are utilized as inputs for our Multi-Regional Input-Output (MRIO) Model. we would like to emphasize that our study is based on

an MRIO modeling approach rather than following a process-by-process approach, which allows us to estimate the carbon footprint reduction potential of autonomous electric vehicles by considering the interdependencies between different sectors and regions. For instance, MRIO model yields the accumulation of CO₂, CH₄, N₂O emissions associated with one-million Euro investment in the automobile manufacturing sector, and we take a portion of it based on the investment made based on the prices provided in the Supplementary Information file, rather than necessitating following an approach which examines each process in the manufacturing of a car part-by-part. While we understand your interest in specific details such as materials, energy, and emissions associated with the cars and their components, the use of MRIO modeling yields an accumulated output for the impacts of manufacturing a car rather than providing the materials, energy, and emissions for the components. There are advantages and disadvantages of using MRIO based models. For more details, please see the following reviews study where we explain the advantages of using MRIO based models for sustainability analysis (e.g. carbon footprint accounting) for electric vehicles:

Onat, N. C., & Kucukvar, M. (2022). A systematic review on sustainability assessment of electric vehicles: Knowledge gaps and future perspectives. Environmental Impact Assessment Review, 97, 106867.

Regarding the economic feasibility of autonomous driving/vehicles, this would be a completely new study requires some different approaches, which would shift the scope of this study significantly.

8. The Table 2 in the SI file, only provide a detailed description of autonomous vehicles, and does not provide a list of materials, energy, and pollution emissions involved.

The explanation for this comment is identical to the previous one. Please see our previous comment.

9. How is the carbon emissions generated in the manufacturing process of components and materials not included in the existing GREET database determined?

The manufacturing processes emissions are determined using MRIO model rather than GREET model. Kindly refer to the Developing Multi-Regional Input-Output model section in the methods section for more information about the model. In our study, we utilized GREET database to determine detailed material composition of the vehicles for end-of-life calculations purposes as we need to determine the vehicle's accumulated materials from the vehicle itself, battery, and autonomous system components in order to estimate the saving in emissions when we recycle such materials compared to emissions emerging from mining those materials. As highlighted earlier, the prices of the vehicles and components are used as inputs for our MRIO model, where we can translate this investment value within each relevant sector to emissions relevant to such activities making the GREET database redundant in determining such impacts for our manufacturing processes. MRIO based approaches used producer prices as proxy to estimate the environmental impacts.

10. The supporting information only investigated the rebound effect in one region of Qatar through a questionnaire. However, in the main text, the authors compared nine countries without indicating whether these countries also experienced a rebound effect.

Thank you for your insightful observation. In our case study, autonomous cars are driven in Qatar, therefore, operational phase related rebound effects are due to the drivers in Qatar. Because there is no car manufacturing in Qatar, these cars have to be manufactured somewhere around the world. Because this is a new emerging technology (there is not commercially available fully autonomous vehicle), producing fully autonomous vehicle involves uncertainties. Taking into consideration the nine countries is to account for the uncertainties involved within such emerging technologies where different countries use different manufacturing processes & technologies, electricity generation mix, and raw materials extraction methods. By doing this, we aim to capture the differences relevant to manufacturing such emerging technologies from a count of the world's biggest automobile manufacturers. These countries also experience rebound effects through manufacturing of components. We added the following text to further clarify: *"This analysis considers a set of potential combinations of manufacturing and assembling sedan*

vehicle bodies, Li-Ion batteries, and autonomous system components in nine different countries represented by twelve sedan models from the world's most renowned automobile manufacturers. The results shown in this study represent an average value of the carbon footprint for each of the twelve car brands considered. Considering multiple combinations aids in assessing the carbon footprint relevant to different choices within various aspects, such as cost, fuel efficiency, and environmental impact."

The manufacturing countries are also affected by this rebound effect. For example, if autonomy increases vehicle use, more parts will be subject to depreciation, wearing, etc. and these parts will be manufactured in the country of origin (manufacturing country) and they will experience increased environmental impacts. This impact can be seen as supply-chain related impact. It is an indirect effect of increased use due to rebound effect. However, this is not only for the manufacturing phase. This is true for all the supply-chain related (indirect) activities involved throughout the life-cycle phases of autonomous vehicles. Therefore, we adopted MRIO-based LCSA approach to account for all the indirect effects (embedded in the global supply chains) of rebound effects.

Per suggestion, we explained global supply-chain related (indirect) impacts of rebound effects as follows:

"Estimating and analyzing environmental impacts across global supply chains are extremely challenging with traditional process-based LCA approaches. While process-level LCA approaches are useful for the quantification of environmental impacts from a life-cycle perspective, they have certain limitations when estimating emissions embedded in the supply chain of the processes in LCA due to cut-off criteria when defining the system boundary. On the other hand, input-output (IO) based LCA approaches are powerful for capturing environmental impacts that are embedded in complex global supply chains, thus allowing a more holistic assessment and can eliminate truncation errors due to the cut-off criteria. However, IO-LCA models introduce some uncertainties due to data aggregation at a sectorial level. Input-output-based approaches are likely to yield more accurate results when estimating environmental impacts. Therefore, in this research, we developed a multi-regional input-output-based LCA approach to account for indirect emissions that are embedded in the complex global supply chains of the manufacturing, operation, and end-of-life phases associated with autonomous electric vehicles."

11. Line 456, the reviewer did not find a comprehensive list of total materials used in autonomous electric vehicles in both the main text and supplementary information. Please explain and supplement this information.

Thank you for pointing out this crucial remark. In agreement with the reviewer's concern, we updated the Supplementary information file and included Table 3 as it shows the material compositions for the twelve vehicle models considered in this study. We have also updated the main text and made a reference to this table to ensure the readers could readily access this list of materials. The update in the manuscript is as follows:

"GREET model specifies all materials used in battery electric vehicles and Lithium-Ion batteries. For the autonomous system components, the material composition of each component is estimated by breaking it down into its subcomponents. Then, aggregating materials coming from the previously mentioned three parts (vehicle body, lithium battery, and autonomous system components) to generate a list of total materials used in autonomous electric vehicles shown in table 3 in the SI"

12. There is no specific list in the supporting information for the manufacturing of the vehicle body, lithium-ion batteries, and autonomous system components. How is the overall carbon emissions of autonomous vehicles calculated?

For each of the three main components of autonomous vehicles (vehicle body, Li-Ion batteries, and Autonomous system components) we have an estimation for how much each of those components cost (producer prices), then this cost is expressed in Million-Euros. Next, we used our MRIO model to determine the impacts associated with each sector that these components are manufactured within. For instance, we used our MRIO model to determine the CO₂, CH₄, N₂O emissions associated with one-million euro economic output of automobile manufacturing sector

for each of the nine countries considered, then we determined the impacts of manufacturing the vehicles by considering the proportion of this investment relevant to one million-euro investment. Kindly refer to equations 1, 2, and 3 in the manuscript for detailed explanations of the MRIO model calculations. We did this for each of the vehicle body, Li-Ion battery, and autonomous system components (where not all components are manufactured within the same sector, so each component was handled using its relevant sector).

13. How are carbon emissions calculated during the autonomous vehicle usage phase? Specifically, what parts of the battery usage are included? In line 459 of the supporting information, why is the carbon emissions calculation of electricity generation using the electricity price?

The carbon emissions within the usage phase are calculated using the MRIO model we developed. Refer to equations 1, 2, and 3 in the Developing Multi-Regional Input-Output Model section in the manuscript. Emissions within this phase stem mainly from impacts associated with charging the Li-Ion batteries and maintenance operations. Since electric vehicles are associated with no tailpipe emissions, they have zero Tank-to-Wheel emissions. As for their Well-to-Tank emissions, they are a representative of all processes carbon emissions involved in charging these batteries. For instance, for the case of Qatar where Natural gas is the main source of electricity generation, the Well-to-Tank operations for an electric vehicle starts from extracting the natural gas, refining, transportation, electricity generation using natural gas, and transmitting electricity to point-of-use. The electricity price is used as an economic output whereby using fuel efficiency (kWh/Km) we estimated the carbon emissions associated with driving the vehicle for one kilometer and multiplied this value by total life cycle mileage to get life cycle Well-to-Tank emissions. As for maintenance, we have used real world data from a local source in Qatar which indicates the maintenance cost for electric vehicles.

14. Autonomous vehicles also consume corresponding materials and energy and produce pollution emissions during the recycling. How did the authors handle the carbon emissions generated from this part in their calculations?

Due to data absence from the MRIO model (EXIOBASE) for recycling impacts for the materials used in the vehicles, we have utilized the literature for Global Warming Potential impacts associated with each of the materials used (expressed in $\text{KgCO}_2\text{-Eq./ Kg material recycled}$). We have also referred to the recovery rate values for the materials from the literature (expressed in %, used to account for material losses during the recycling processes. Using the accumulated materials list mentioned in comment #11, lets say a certain material has an amount of X in the vehicle, we divided X by the recovery rate to get the actual amount that needs to be recycled, undergo some material losses to get the final amount of X, then we multiplied $X/\text{Recovery rate}$ by the GWP values to get the total GWP for a certain material in the vehicles. Lastly, we have accumulated the total GWP for each material present in the vehicles, to get the total recycling GWP per vehicle.

We have then calculated the carbon emissions associated with mining the materials as their total value in the vehicles are estimated by multiplying the amount present (in Kg) by the price ($\$/\text{Kg}$) to get how much it would cost to mine those materials. Then we used this cost as an investment value expressed in Million-Euro and used them as inputs for the MRIO model, where each material was handled with its relevant mining sector. This step yields the GWP for mining those materials, which are accumulated to have the total mining GWP per vehicle. In order to determine the saving in emissions due to recycling, we simply subtract the total recycling GWP per vehicle from the total mining GWP per vehicle to get how much we would be saving in emissions when we recycle rather than mining for materials all over again. We presented the calculation process in the manuscript as follows:

“GREET model specifies all materials used in battery electric vehicles and Lithium-Ion batteries. For the autonomous system components, the material composition of each component is estimated by breaking it down into its subcomponents. Then, aggregating materials coming from the previously mentioned three parts (vehicle body, li-ion battery, and autonomous system components) to generate a list of total materials used in autonomous electric vehicles shown in table 3 in the SI. Afterwards, we utilized the MRIO model to account for the mining emissions of each material relevant to its amount within the vehicle. As for the recycling process, Due to data absence from the MRIO model for recycling impacts for the materials used in the vehicles, we have utilized the literature for Global Warming Potential impacts associated with each of the materials used (expressed in $\text{KgCO}_2\text{-Eq./ Kg material recycled}$). Using the total materials list for each material, we accounted for the recovery rate to get the actual amount

that needs to be recycled, as within recycling processes there would be some material losses. Lastly, we have accumulated the total GWP for each material present in the vehicles, to get the total recycling GWP per vehicle, which would be subtracted from the total mining GWP per vehicle to get how much the saving on emissions would be in case of recycling.”

15. There are some issues with the reference citations in the main text and supporting information.

We used Mendeley referencing tool and sometimes when converting from docx to .pdf, some issues happen, and some references become unavailable with broken links. We corrected this issue in the revised version. Thank you for bringing this to our attention.

16. The source of the data in Table 4 of the supporting information is unknown.

The Supplementary information file has been updated to include a brief description of Table 4 and the source of the data presented in the SI. The source of the data is the survey conducted, as well as the bootstrapping simulation. Kindly refer to section 2.1 Annual Travelling Distances Analysis in the SI for more information on this topic. About the bootstrapping simulation, we have explained it in that section as follows: “Due to the limited amount of data points of 330, Bootstrapping was done on the three sets of minimum, average, and maximum rebound-adjusted annual traveling distances by using sampling with replacement. Bootstrapping involved resampling a single dataset to produce a massive number of simulated samples. For each of the three sets, 330 values of rebound-adjusted annual traveling distances are sampled with replacement from the original rebound-adjusted annual traveling distance sets of 330 values each. The generated 330 values form one sample. Then, the average of the generated sample was calculated. This procedure was repeated to generate 10,000 sample averages using Python. Fig. 3 summarizes our bootstrapping results shown within a box and whisker plot. These three sets are of vital importance due to their use in estimating the rebound effect as well as estimating service life for autonomous vehicles. Furthermore, the bootstrapping procedure was also done on the original minimum, average, and maximum annual traveling distances to estimate the service life of battery electric vehicles. Then, the difference in the service life between autonomous and non-autonomous vehicles is used to estimate a percentage decrease in service life due to the rebound effect. This percentage represents the rebound effect which is calculated as in Equation(5).”

The following explanation is added to the SI:

“The following table summarizes the annual travelling distances sets. The minimum, average, and maximum annual travelling distances are extracted from the survey, where each reading reflects the participant’s annual travelling distances values. As for minimum, average, and maximum annual travelling distance sample averages, they were extracted from carrying our bootstrapping simulation on the three original distances sets extracted from the survey. Lastly for minimum, average, and maximum rebound-adjusted annual travelling distance sample averages, we took the three original distances set, adjusted them with the rebound effect to reflect the increased annual travel due to autonomy, and then applied bootstrapping simulation.”

17. Lines 315 to 318 of the supporting information, the relevant calculations mainly focus on prices, and the average values of technology, raw materials, and energy for the nine countries are not reflected.

As mentioned before, the study followed an MRIO model approach, which accumulates impacts based on economic output of a sector rather than conducting a process-by-process analysis. This modeling technique requires an economic output for each sector which is the reason of the calculations to estimate the costs of certain components in countries where relevant data could not be found. This estimation is based on the ratio between the GDP of a country with available data and the country with missing data. This approach helps provide insights into the potential costs associated with the components in question. Furthermore, our study clarifies that the MRIO model used in the analysis specifically focuses on carbon emissions. As mentioned earlier in comment #10, the choice of nine countries in the analysis aimed to investigate the uncertainties related to autonomous vehicles, which is an emerging technology. The energy mix and material requirement of the producer countries are already reflected in our model through the MRIO modeling. In other words, the MRIO model have a matrix called “requirement matrix” which

represents the input requirement of a sector to produce an output. These calculations are explained in the manuscript as follows:

“Estimating and analyzing environmental impacts across global supply chains are extremely challenging with traditional process-based LCA approaches. While process-level LCA approaches are useful for the quantification of environmental impacts from a life-cycle perspective, they have certain limitations when estimating emissions embedded in the supply chain of the processes in LCA due to cut-off criteria when defining the system boundary⁴⁰. On the other hand, input-output (IO) based LCA approaches are powerful for capturing environmental impacts that are embedded in complex global supply chains, thus allowing a more holistic assessment and can eliminate truncation errors due to the cut-off criteria⁴¹. However, IO-LCA models introduce some uncertainties due to data aggregation at a sectorial level. Input-output-based approaches are likely to yield more accurate results when estimating environmental impacts⁴². Therefore, in this research, we developed a multi-regional input-output-based LCA approach to account for indirect emissions that are embedded in the complex global supply chains of the manufacturing, operation, and end-of-life phases associated with autonomous electric vehicles.”

AND

“Multi-Regional Input-Output (MRIO) Modeling is an economic technique that keeps track of money transfers among the main economic sectors across major economies in the world. Environmentally extended MRIO modeling allows to track and analyze resource flows by incorporating information from the National Footprint Databases and Biocapacity Accounts. MRIO models allow to track of resource flows between a country's primary economic sectors and satellite national accounts and breaking down national Footprint data into more targeted consumption- and industry-related components⁴⁴. Global financial transactions between governments and regions can be captured using MRIO models because of this, MRIO analysis can help to analyze how economic activity in certain sectors and countries can cause environmental impacts with consideration of global supply chains⁴⁵. Constructing the MRIO model using Leontief's equation in equation (1) to access implications associated with a specific sector's output unit as well as unintended consequences of the industry's international supply networks.

$$X = (I - A)^{-1} \times y \quad (1)$$

Where X: Total output column vector (M€); I: identity matrix; A: Coefficient matrix (M€/M€); y: total demand column vector (M€).

Next, equation (2) is used to obtain multipliers indicating the impacts of a given sector's input of one million Euros.

$$B = E \times (\text{DIAG}(X))^{-1} \quad (2)$$

Where B: Matrix of intensities (Per M€); E: Environmental satellite accounts

Finally,

$$r = B \times L \times y \quad (3)$$

Where r: total output of each sector vector; B: intensity matrix per unit of output; L:

In this study, EXIOBASE 3.8.2 is used as the MRIO database⁴⁶. The Supply and Use Tables are used to create EXIOBASE's MRIO datasets, which comprise national and global input-output tables as well as raw data from the UN's System of National Accounts, Comtrade, and Eurostat databases, all at current prices with a steady product sales assumption. The environmental impacts considered in this study are emissions of Carbon dioxide (CO₂), Methane (CH₄), and Nitrogen dioxide (N₂O). Then, using these emissions, Global Warming Potential (GWP) is estimated based on a 100-years' time series as per IPCC's 6th assessment report⁴⁷. GWP is a measure of a greenhouse gas's ability to trap energy over a given time. This amount of energy absorbed by a ton of a specific GHG is compared to a ton of CO₂'s ability to trap energy.”

Please see the following studies for more detail about input-output based LCA approaches:

Joshi, S. (1999). Product environmental life-cycle assessment using input-output techniques. Journal of industrial ecology, 3(2-3), 95-120

Wiedmann, T. (2009). Carbon footprint and input-output analysis—an introduction.

Nakamura, S., & Nansai, K. (2016). Input-output and hybrid LCA. Special types of life cycle assessment, 219-291.

Hendrickson, C. T., Lave, L. B., & Matthews, H. S. (2010). Environmental life cycle assessment of goods and services: an input-output approach. Routledge.

Onat, N. C., Kucukvar, M., Halog, A., & Cloutier, S. (2017). Systems thinking for life cycle sustainability assessment: A review of recent developments, applications, and future perspectives. Sustainability, 9(5), 706

Reviewer #2 (Remarks to the Author):

The manuscript Rebound Effects Undermine Carbon Footprint Reduction Potential of Autonomous Electric Vehicles presents comprehensive analysis. However, manuscript can only be published after following major revisions.

1. There are numerous places where references are missing.

We used Mendeley referencing tool and sometimes when converting from docx to .pdf, some issues happen, and some references become unavailable with broken links. We corrected this issue in the revised version. Thank you for bringing this to our attention.

2. The authors used GREET model to estimate material composition in vehicles. The authors need to explain the suitability of the GREET model for other nations, it is important to consider the model's applicability to different contexts. The GREET model was initially developed for the United States and focuses on vehicle technologies and energy pathways specific to the U.S. Therefore, when applying the GREET model to other nations, it may require adjustments and customization to accurately reflect the local energy mix, transportation infrastructure, and vehicle characteristics.

We appreciate your insightful comment. In our study, we utilized the GREET model to extract material composition data. It is important to note that the GREET model does not provide specifications for a specific car model in the United States, but rather offers specifications for a generic car model using various data sources including literature reviews, vehicle tear-down data, and vehicle models (1). An updated for their model released in 2020 has stated that they utilized three midsize cars, four small SUVs, and three pickup trucks that serve as high market share exemplars of the US light duty fleet, then used the average material compositions in their model (2). Consequently, the slight disparities in manufacturing processes among various car brands originating from different countries are unlikely to significantly impact our findings. This is due to the fact that the GREET model encompasses characteristics applicable to a generic car model, rather than being tailored to a specific one. Furthermore, it is crucial to highlight that we did not employ the energy requirements obtained from the GREET model for our analysis. Instead, we solely extracted material composition data from this model. Our focus on material composition rather than energy requirements stems from the nature of our research, which specifically used for end-of-life considerations. Consequently, the GREET model was solely utilized for the calculation of the End-of-Life phase, which, as evidenced by our sensitivity analysis, was determined to have the least influential impact on the life cycle Global Warming Potential.

(1): <https://greet.es.anl.gov/files/update-veh-specs>

(2): https://greet.es.anl.gov/files/vmc_2020

3. The manuscript needs better organization. A lot of important things such as data sources and uncertainty analysis

are provided in supplementary information. A part of that information in the form of tables or figures should be included in the main manuscript.

We appreciate your observation regarding the organization of the manuscript. In agreement, we made improvements to enhance the clarity and accessibility of important information. Specifically, we have included a table in the main manuscript containing critical values for our study, including battery capacities, fuel efficiency, vehicle body production cost, battery production cost, and autonomous system components production cost for the twelve models considered. Furthermore, we have consolidated the sensitivity analysis results for all four scenarios within the main manuscript, instead of limiting it to only one case. These changes aim to provide a comprehensive and coherent presentation of the data, addressing the concern you raised about supplementary information.

Reviewer #3 (Remarks to the Author):

The paper provides well-defined and impactful objectives. It investigated tradeoffs between improved fuel economy and rebound effects from a life-cycle perspective.

Several comments are provided:

- The life cycle operations with quantitative results obtained, the calculations and scope is based on a product, local, regional or global study, further supplement would allow readers with a clearer understanding on the paper.

The study quantifies global impacts of local use. So, it means autonomous vehicles use phase are in Qatar, thus survey represents the population (people who can drive only) in Qatar. Furthermore, while the use phase in Qatar and the majority of use phase impacts are inside Qatar. However, we also captured supply-chain linked impacts associated with use phase. Kindly refer to the Developing Multi-Regional Input-Output model section in the manuscript for more details about this method. It is a comprehensive method and covers impacts associated with global supply chains. However, manufacturing phase and end-of life phases took place outside Qatar, as there is no car manufacturing and proper end-of-life management facilities for car recycling in Qatar. In the manuscript, these aspects are explained as follows:

*“Estimating and analyzing environmental impacts across global supply chains are extremely challenging with traditional process-based LCA approaches. While process-level LCA approaches are useful for the quantification of environmental impacts from a life-cycle perspective, they have certain limitations when estimating emissions embedded in the supply chain of the processes in LCA due to cut-off criteria when defining the system boundary(Suh et al., 2004). On the other hand, input-output (IO) based LCA approaches are powerful for capturing environmental impacts that are embedded in complex global supply chains, thus allowing a more holistic assessment and can eliminate truncation errors due to the cut-off criteria(Ward et al., 2018). However, IO-LCA models introduce some uncertainties due to data aggregation at a sectorial level. Input-output-based approaches are likely to yield more accurate results when estimating environmental impacts(Pomponi & Lenzen, 2018). **Therefore, in this research, we developed a multi-regional input-output-based LCA approach to account for indirect emissions that are embedded in the complex global supply chains of the manufacturing, operation, and end-of-life phases associated with autonomous electric vehicles.** The functional unit of the analysis is per vehicle. The life cycle phases and processes involved in the analysis are presented in Fig. 6.*

Fig. 6. System boundary for life cycle assessment of autonomous vehicles”

- About the rebound effects and whether AVs have fewer emissions, what is the theoretical support for the causal loop diagram model (CLD) proposed in lines 132-138 of the text? What are the model's design sources in the absence of relevant literature support? Line 153 of the text reinforces loops 4 and mentions expected fuel efficiency improvements. Is there any supporting data? Line 170 mentions rebound effects and behavioral factors leading to the environmental impacts of autonomous electric vehicles. Can a brief list of the influencing factors be given?

About lines 132-138 (now 171-177 after revision) we have cited a research paper supporting each interaction with the CLD modeling. By doing this, we strengthen our model ensuring it accurately identifies interrelations between the different elements. In this causal loop diagram, we do not claim autonomous vehicles have fewer or more emissions compared to non-autonomous vehicles. We are rather explaining a mechanism in which these interactions between each variables are explained in the relevant studies cited in the manuscript. For example, variables autonomy and fuel efficiency have positive causation as increasing one would increase the other, while decreasing one the other would also exhibit the same behavior. This relationship was obtained from the literature (Mersky & Samaras, 2016).

About line 153 (now 192 after revision) we have cited one research paper that supports this statement. However, for the amount of this improvement, we utilized a value from one research paper that indicates that autonomous vehicles have the potential to enhance fuel efficiency by up to 23.6% (Kavas-Torris et al., n.d.).

About line 170 (now 210 after revision) we have made this section clearer as we clearly listed all the behavioral factors, which we investigated as mentioned here: “The behavioral factors investigated include age, marital status, employment, level of education, income, number of adults per household, number of cars per household, and driving experience in years”. We intend to investigate which certain groups are more likely to travel more annually compared to people not conforming to such groups. For instance, it could be the fact that married people with children travel more compared to single people. This analysis aids in identifying such groups to help policymakers make informed decisions about adopting autonomous vehicles for different groups of people. Please refer to Behavioral Analysis section in the manuscript for more information about this topic.

- About the carbon footprint of EV, four scenarios from production to use to end of life according to the LCA assessment methodology are provided. This gives the conclusions of the analysis. However, line 235 refers to 12 vehicle types, which ones are they, and whether the LCA was carried out on all the vehicle components or whether it is suggested that this section could be described in more detail. Line 274 refers to renewable energy, whereas renewable energy should refer to Scenario 3 and Photovoltaics in Scenario 3 and 4. It is suggested that this be clarified.

The twelve mentioned vehicle types represent some of the world's most renowned car manufacturer's Sedan models. We have considered nine countries which are represented by twelve car brands. We believe this would aid in investigating uncertainties associated with autonomous vehicles as this approach would utilize the analysis for all brands having different fuel efficiencies, manufacturing technologies, and electricity generation mix. The results show the average value for the carbon footprint of the twelve brands considered. By doing so, considering all possible combinations (e.g, manufacturing of major car components such as car shell, battery, autonomous system in different countries) aids in assessing the carbon footprint relevant to different choices within various aspects, such as cost, fuel efficiency, and environmental impact. We have adjusted the relevant section in the manuscript after the causal loop diagram modeling to contain a brief explanation of the consideration of the different brands. In line 274 (now 317 after revision) we discuss the results of Scenario 3 and 4 where Photovoltaics as an example of renewable energy is utilized to charge the batteries of non-autonomous and autonomous battery electric vehicles, respectively. We have revised this part and ensured readers would be conveniently able to be aware of this fact.

- The discussion section describes the results of the analysis of the entire study. Furthermore, based on the results, suggestions are made for the future development of AVs. However, the article's title considers the future development of AVs from the perspective of the rebound effect, and the discussion section should suggest ways to mitigate the increase in carbon emissions due to the rebound effect. The CLD cycle model mentioned in the previous section should also be discussed as appropriate. The sensitivity analysis results are mentioned in line 349, but more details of the results could be presented.

Thank you for your insightful comment. The discussion section of the article acknowledges the importance of addressing the rebound effect in future development of AVs. It has been updated to suggest ways to mitigate the increase in carbon emissions specifically related to the rebound effect, including the need for policies targeting end-of-life phase emissions, fuel efficiency improvements, rebound effect mitigation, and sustainable manufacturing practices. Additionally, the CLD cycle model is appropriately discussed, explaining the reinforcing loops and balancing loops of rebound effects. While the sensitivity analysis results are briefly mentioned, providing more details of the results as we believe it would enhance the discussion and offer a comprehensive understanding of their implications in light of your comment.

- In the questionnaire studying the behaviour. Further details could be provided including the source of the questionnaire, reliability analysis (calculating Cronbach's alpha), and validity analysis results of the questionnaire.

A stated preference survey was designed for this paper as a web-based survey questionnaire was developed which comprises of collecting respondents' socio-economic information (such as age, education, employment status, driving experience, and their belief and attitudes towards AVs technology). The questionnaire was designed in Arabic and English languages using the Qualtrics platform, which is a web-based survey tool. The questionnaire comprised three different sections. The first section includes questions regarding socio-demographic characteristics of the respondents such as age, ethnicity, average income, educational status, occupational status. The second section included some questions related to commuting behaviors. In the third section, questions related to individuals' knowledge about AV, perceptions of AV's safety, performance in harsh environmental conditions, security, travel time, congestions, comfort, and operational costs, and their preference of shifting to AV. A brief list of the questions and the possible answer choices are provided in the SI file in Section 2. Most of the questions in the third section were not utilized by this study as it is not included in our scope to assess people's perception of AVs in terms of what they think about their benefits, drawbacks, and possibility of switching to AVs when they become available. Participants were provided with a brief introduction of the AV technology prior to answering the questionnaire as follow: "An autonomous vehicle (AV) is basically a vehicle which can guide itself with no need of human control. In other words,

autonomous vehicles are known as self-driving cars, where the computer will take the lead for driving. Moreover, autonomous vehicles use different kinds of technology. They can be connected with the Global Positioning System (GPS) to help with navigation, where the car will display information to users to choose his/her destination. Furthermore, the autonomous vehicle will be provided with sensors and other tools to avoid collisions”.

An approval was obtained from Qatar University’s ethical committee (QU-IRB) before distributing the questionnaires. The questionnaire was spread through information leaflets posted via different networking organizations and social media platforms. In total 589 respondents filled the questionnaire, of which 443 filled it out in English while 146 filled it out in Arabic. The data was filtered for the respondents of age at least 18 years (legal age limit qualifying for driving license), missing income data, missing rebound-effect related question data, and missing data related to AV background knowledge. In this regard, 9, 145, 104, 1 respondents were eliminated from the sample, respectively. Thus, 330 respondents were considered as a final sample for the analyses.

As for the reliability analysis, we have concluded with Cronbach’s alpha value of 0.72 after excluding 21 variables out of the original 35 variables in the survey. Some studies, such as (Hair, 2009; Moss et al., 1998; Ursachi et al., 2015), lend credence to the notion that a Cronbach alpha value surpassing 0.6 is deemed satisfactory, especially for small-sized samples. It is noted that we only utilized a couple of the variables for our life cycle carbon emissions calculations in this study which were the annual travelling distance and the rebound effect variables which were both included in the 14 variables used for calculating this value. Some of the excluded variables were utilized in the proportion odd models in the behavioral analysis including background knowledge of AVs, age, and marital status.

Thank you for your time and insightful feedback.

References

)+ والتواصل النقل و احصاءات النقل CHAPTER X TRANSPORT AND COMMUNICATIONS STATISTICS. (n.d.).

Hair, J. (2009). *Multivariate Data Analysis. Faculty and Research Publications.*
<https://digitalcommons.kennesaw.edu/facpubs/2925>

Kavas-Torris, O., Cantas, M. R., Aksun-Guvenc, B., Guvenc, L., & Cime, K. M. (n.d.). *The Effects of Varying Penetration Rates of L4-L5 Autonomous Vehicles on Fuel Efficiency and Mobility of Traffic Networks ADAS enhanced Powertrain Controls View project The Effects of Varying Penetration Rates of L4-L5 Autonomous Vehicles on Fuel Efficiency and Mobility of Traffic Networks.*
<https://doi.org/10.4271/2020-01-0137>

Mersky, A. C., & Samaras, C. (2016). Fuel economy testing of autonomous vehicles. *Transportation Research Part C: Emerging Technologies*, 65, 31–48. <https://doi.org/10.1016/J.TRC.2016.01.001>

Moss, S., Prosser, H., Costello, H., Simpson, N., Patel, P., Rowe, S., Turner, S., & Hatton, C. (1998). Reliability and validity of the PAS-ADD Checklist for detecting psychiatric disorders in adults with intellectual disability. *Journal of Intellectual Disability Research : JIDR*, 42 (Pt 2)(2), 173–183.
<https://doi.org/10.1046/J.1365-2788.1998.00116.X>

Pomponi, F., & Lenzen, M. (2018). Hybrid life cycle assessment (LCA) will likely yield more accurate results than process-based LCA. *Journal of Cleaner Production*, 176, 210–215.
<https://doi.org/10.1016/j.jclepro.2017.12.119>

- Suh, S., Lenzen, M., Treloar, G. J., Hondo, H., Horvath, A., Huppes, G., Jolliet, O., Klann, U., Krewitt, W., Moriguchi, Y., Munksgaard, J., & Norris, G. (2004). System Boundary Selection in Life-Cycle Inventories Using Hybrid Approaches. *Environmental Science & Technology*, 38(3), 657–664. <https://doi.org/10.1021/es0263745>
- Ursachi, G., Horodnic, I. A., & Zait, A. (2015). How Reliable are Measurement Scales? External Factors with Indirect Influence on Reliability Estimators. *Procedia Economics and Finance*, 20, 679–686. [https://doi.org/10.1016/S2212-5671\(15\)00123-9](https://doi.org/10.1016/S2212-5671(15)00123-9)
- Verma, J. P., & Verma, P. (2020). Determining Sample Size and Power in Research Studies. *Determining Sample Size and Power in Research Studies*. <https://doi.org/10.1007/978-981-15-5204-5>
- Ward, H., Wenz, L., Steckel, J. C., & Minx, J. C. (2018). Truncation Error Estimates in Process Life Cycle Assessment Using Input-Output Analysis. *Journal of Industrial Ecology*, 22(5), 1080–1091. <https://doi.org/10.1111/JIEC.12655>

REVIEWERS' COMMENTS

Reviewer #1 (Remarks to the Author):

The authors have properly answered the reviewers' comments. The author's efforts in revising the manuscript are commendable and the manuscript is recommended for publication.

Reviewer #2 (Remarks to the Author):

The authors have made revisions to the manuscript as per the feedback provided in the first round of reviews. I am happy with the changes. The manuscript can be accepted for publication.

Reviewer #3 (Remarks to the Author):

Thanks for the replies to the comments and the corresponding update. There is no further comments on the manuscript.

Detailed Response to Reviewers' Comments

The authors are deeply appreciative of the reviewers' and editor's invaluable feedback. All reviewer suggestions have been carefully addressed and incorporated into the manuscript, with the modifications clearly indicated in blue for their reference.

Reviewer #1 (Remarks to the Author):

The authors have properly answered the reviewers' comments. The author's efforts in revising the manuscript are commendable and the manuscript is recommended for publication.

We would like to express our sincere gratitude for your valuable feedback and insightful comments. Thank you for your time and expertise in reviewing our work.

Reviewer #2 (Remarks to the Author):

The authors have made revisions to the manuscript as per the feedback provided in the first round of reviews. I am happy with the changes. The manuscript can be accepted for publication.

We would like to express our sincere gratitude for your valuable feedback and insightful comments. Thank you for your time and expertise in reviewing our work.

Reviewer #3 (Remarks to the Author):

Thanks for the replies to the comments and the corresponding update. There is no further comments on the manuscript.

We would like to express our sincere gratitude for your valuable feedback and insightful comments. Thank you for your time and expertise in reviewing our work.